# A cercarial invadolysin interferes with the host immune response and facilitates infection establishment of *Schistosoma mansoni*

**Jacob R. Hambrook, Patrick C. Hanington** *

School of Public Health, University of Alberta, Edmonton, Alberta, Canada

* pch1@ualberta.ca

**Data Availability Statement:** All relevant data are within the paper and its Supporting Information files.

## Abstract

*Schistosoma mansoni* employs immune evasion and immunosuppression to overcome immune responses mounted by its snail and human hosts. Myriad immunomodulating factors underlie this process, some of which are proteases. Here, we demonstrate that one protease, an invadolysin we have termed SmCI-1, is released from the acetabular glands of *S. mansoni* cercaria and is involved in creating an immunological milieu favorable for survival of the parasite. The presence of SmCI-1 in the cercarial stage of *S. mansoni* is released during transformation into the schistosomula. SmCI-1 functions as a metalloprotease with the capacity to cleave collagen type IV, gelatin and fibrinogen. Additionally, complement component C3b is cleaved by this protease, resulting in inhibition of the classical and alternative complement pathways. Using SmCI-1 knockdown cercariae, we demonstrate that SmCI-1 protects schistosomula from complement-mediated lysis in human plasma. We also assess the effect of SmCI-1 on cytokine release from human peripheral blood mononuclear cells, providing compelling evidence that SmCI-1 promotes an anti-inflammatory microenvironment by enhancing production of IL-10 and suppressing the production of inflammatory cytokines like IL-1B and IL-12p70 and those involved in eosinophil recruitment and activation, like Eotaxin-1 and IL-5. Finally, we utilize the SmCI-1 knockdown cercaria in a mouse model of infection, revealing a role for SmCI-1 in *S. mansoni* survival.

## Author summary

Schistosomiasis continues to afflict over 230 million people globally. The parasitic flatworms serving as the causative agents of this disease produce numerous factors to facilitate infection and survival in a human host. This is especially true during initial infection, where they penetrate through human skin in search of entry into the circulatory system. During migration through the skin, they must overcome the immune response. To do so, they produce numerous immunosuppressive factors. Here, we functionally characterize one such factor, a *S. mansoni* cercarial invadolysin (SmCI-1) that is released from penetrating cercaria during their transformation into schistosomula. We demonstrate that SmCI-1 aids in the degradation of both structural and immunologically-relevant proteins

**Funding:** This work was supported by funding from the Natural Sciences and Engineering Council of Canada (NSERC) grants 2018-05209 and 2018-522661 to PCH. The funders had no role in study design, data collection and analysis, decision to publish, or preparation of the manuscript.

**Competing interests:** The authors have declared that no competing interests exist.

found in the skin and show that the cleavage of complement component C3b reduces complement mediated lysis of the developing parasite. We also show that treatment of human leukocytes with SmCI-1 reduces the release of key immunostimulatory cytokines in the presence *S. mansoni*-derived factors and LPS. Knockdown of SmCI-1 in cercaria reduces infection success and adult worm burden in a mouse model system.

## Introduction

Despite significant investment into treatment and prevention programs, schistosomiasis continues to afflict an estimated 230 million people worldwide [1]. Key among the numerous ways in which schistosomes persist in human populations are immunomodulatory mechanisms that interfere with and suppress the immune response of their gastropod and human hosts [2–4]. While the strategies used to overcome immune responses are highly effective, they are not perfect. Schistosome-infected individuals (and animals) can develop low levels of non-sterilizing immunity to infection, and can produce immune factors capable of killing the larval worms [5–8]. Such findings demonstrate that an in-depth examination of the survival mechanisms at various stages of the intra-human schistosome life cycle merits investigation as possible targets for novel therapeutics and/or vaccine development efforts.

Proteases are prominently featured amongst the immunomodulatory mechanisms employed by *Schistosoma mansoni* during the snail and human infection process [9–11]. Various life cycle stages employ different types of proteases to several different ends, including degradation of host structural and immunological components, while also interfering with coagulation [10–18]. Proteases are especially important during cercarial penetration of human skin, where both mechanical disruption of host cell-to-cell interactions as well as the release of proteases from the acetabular glands help facilitate movement of the parasite towards a host venule [15,19–21]. The release of such factors from the parasite during penetration has been demonstrated to facilitate larval survival [22]. In *S. mansoni*, the most abundant protein released from these acetabular glands is a 28/30 kda serine protease termed *S. mansoni* cercaria elastase (SmCE) that makes up ~36% of the contents of the acetabular glands [23]. This protein cleaves key structural components in skin such as elastin, laminin, fibronectin, collagen type IV, and keratin, as well as complement component C3, highlighting a role in overcoming both structural and immune barriers in the skin [24–27]. Broad inhibition of serine proteases, as well as SmCE specifically, substantially reduces cercarial penetration success, emphasizing the essential function of SmCE during the initial stages of *S. mansoni* infection of their human host [27,28].

Also present in cercaria are several metalloproteases that have yet to be functionally characterized [23,29–31]. Key among the metalloproteases found in *S. mansoni* cercaria are a subset of matrix metalloproteases (MMPs) known as invadolysins, which are known immunomodulators [32–36]. Invadolysins are one of the most expanded gene families in schistosomes, and homologous invadolysins are present in all three of the predominant schistosomiasis-causing species [31,37–41]. Seven invadolysins have been shown to be differentially regulated in *S. mansoni* to date, with five being most highly transcribed in the germ ball stage of development, during which the necessary products for skin penetration by cercaria are generated [31]. The other two are most highly upregulated in cercaria themselves [31]. The predominant generation of invadolysins during intramolluscan development of cercaria has been confirmed in avian schistosomes such as *Trichobilharzia szidati*, where emerged cercaria were found to feature high levels of invadolysin protein while transcript levels remained low [42]. Invadolysins

are also found in *S. japonicum* cercaria, with the three upregulated invadolysin genes observed in an *S. japonicum* micro-array bearing high levels of amino acid similarity to three of the most highly upregulated invadolysins in *S. mansoni* [40]. The case for characterization of invadolysins during initial infection of a human host is bolstered by the observation that the second most prominent invadolysin in *S. mansoni* germ balls (Smp_090100.1) has also been identified as the second most abundant protein released from *S. mansoni* acetabular glands during transformation into schistosomula, comprising roughly 12.8% of the normalized volume of the acetabular glands [23,31]. While Curwen et al. referred to this invadolysin as SmPepM8, for the purposes of this work, we refer to Smp_090100.1 as *S. mansoni* cercarial invadolysin 1 (SmCI-1) in keeping with the nomenclature surrounding cercarial elastase (SmCE).

The utilization of invadolysins as tools for invasion of a host by a parasite has been extensively studied, especially in the case of kinetoplastid parasites such as *Leishmania spp.* and *Trypanosoma spp.* Invadolysins were initially characterized in *Leishmania spp.* and are often termed leishmanolysins, or glycoprotein 63kda (GP63) [43,44]. GP63 possesses numerous functions, including the degradation of key cell structure components, the degradation of relevant immune molecules such as IgG, CD4 and complement component C3, as well as the alteration of immune cell functions such as chemotaxis and ability to generate a respiratory burst response [45–58]. Ultimately, these immunomodulatory roles are species dependent, and are often also dependent upon the obstacles faced by the respective life cycle stage of the parasite [48].

To our knowledge, the only helminth metalloprotease to have been functionally characterized as a known invadolysin is our previous work on a protein termed SmLeish (Smp_135530) that was shown to inhibit the movement of *Biomphalaria glabrata* haemocytes, and to negatively impact the kinetics of *S. mansoni* establishment within *B. glabrata* [13]. Other helminth metalloproteases have been implicated in immunomodulation as well, with *Necator americanus* metalloproteases demonstrating a specific anti-eosinophil mechanism of action via the cleaving of Eotaxin-1 [59].

Given the consistent immunomodulatory role played by invadolysins during parasite infection/establishment, and the presence of SmCI-1 as the second most abundant protein in *S. mansoni* acetabular glands, we set out to characterize SmCI-1 in the context of early *S. mansoni* infection of the human host. In this study, we demonstrate that SmCI-1 acts as a MMP and can cleave specific human proteins relevant to structural and immunological functions. Additionally, we examined the capacity of SmCI-1 to alter cytokine output of human polymorphic blood mononuclear cells. This assessment identified anti-inflammatory properties. To follow up on this observation we used a mutant form of SmCI-1 lacking MMP activity to demonstrate that reduction of key inflammatory cytokines required MMP activity, while eliciting IL-10 production did not. All of this was done with the intent of better understanding how *S. mansoni* is capable of surviving within human skin, as well as gaining a deeper knowledge of how parasitic organisms employ metalloproteases during infection and establishment within a host.

## Methods

### Ethics statement

Swiss-Webster mice were provided by the NIH/NIAID Schistosomiasis Resource Center at the Biomedical Research Institute [60]. All animal work observed ethical requirements and was approved by the Canadian Council of Animal Care and Use Committee (Biosciences) for the University of Alberta (AUP00000057).

## Structure predictions and visualization

Homology modeling was performed by submitting the amino acid sequence for SmCI-1 (Smp_090100.1) to the Robetta Server (robetta.bakerlab.org). RoseTTAFold, a deep learning-based modeling method, was used for secondary/tertiary structure prediction. The prediction with the highest confidence (0.71) was uploaded to PyMol (PyMOL Version 2.4.1, Schrodinger) for visualization and labeling of key active site amino acids. For GP63 comparison, the GP63 protein databank file was obtained from the RCSB protein databank [61].

## Production of recombinant SmCI-1 and purification

The complete mRNA sequence of SmCI-1 was synthesized as cDNA using GenScript's gene synthesis service and inserted into a pHEK293 Ultra Expression Vector I. Transient transfection of HEK293 cells was performed as per the manufacturer's recommendations. Briefly, 30μg of SmCI-1plasmid were added to 60μl of 293fectin Transfection Reagent (Thermo Fisher Scientific) and added to suspension HEK cells for 2 days. The recombinant protein was then purified via the anti-SmCI-1 polyclonal antibody generated against a recombinant SmCI-1 protein produced in a bacterial expression system (see below). A mutant form of SmCI-1 (SmCI-1Mut) was synthesized in an identical manner, with the Glu232 codon (GAA) replaced with a Glycine codon (GGG). This amino acid change was made because Glu232 possesses a carboxyl group which acts as a nucleophile during proteolysis, whereas glycine lacks a side chain, rendering it incapable of facilitating this process. rSmCI-1Mut was purified in an identical manner to the Wild Type (WT) rSmCI-1.

## Production of Rabbit anti-SmCI-1 antibody

The synthesized SmCI-1 sequence was inserted into the pET-47b(+) bacterial expression vector (Novagen) following manufacturers specifications. BL21(DE3) (Thermo Fisher Scientific) cells were transfected with SmCI-1-containing pET-47b(+) vector, grown at 37˚C in LB media (Thermo Fisher Scientific) containing 100μg/ml kanamycin (Thermo Fisher Scientific) and then induced with 1 mM isopropyl β-d-1-thiogalactopyranoside (IPTG) once the cells had grown to OD 600. Following ~4 hours of growth, the cells were concentrated by centrifugation at 10 000 rpm for 20 minutes at 4˚C. The final weight of the bacteria pellet was measured and then the lysing reagent B-PER (Thermo Fisher Scientific) was added at a concentration of 4 ml/g of bacteria in combination with phenylmethylsufonyl fluoride (PMSF, final concentration of 1 mM), mixed gently and left to incubate at room temperature for 15 minutes. Before application to the 5ml 6xHIS column (GE Healthcare) for purification, the supernatant was diluted to a total protein concentration of 100 μg/ml in binding buffer (GE Healthcare). Following purification, the 6x HIS tag region of the recombinant protein was removed following the vector manufacturers specifications (Novagen). This recombinant SmCI-1 was used to generate a rabbit anti-SmCI-1 polyclonal antibody using the Custom pAb service offered by Genscript.

## *S. mansoni* culture

100 M-line *Biomphalaria glabrata* snails were experimentally infected with 10 *Schistosoma mansoni* (NMRI strain) miracidia in 12 well plates for 24 hours, before being raised in artificial spring water (ASW) for 6–8 weeks, at room temperature, on a diet consisting of red leaf lettuce. Snails were allocated into 6 well plates containing ASW and exposed to light for 5 hours during which cercaria emerged. Cercaria were mechanically transformed into schistosomula by vortexing for 60 seconds in 1.5 ml tubes. Parasites were kept on ice for 15 minutes, before

being centrifuged for 5 minutes at 1000g, 4˚C. Artificial spring water was removed, and 200 μl of RPMI 1640 was added. Cercaria were again vortexed for one minute, and visually inspected using microscopy to confirm detachment of their tails. Parasites were centrifuged again, supernatants were collected and frozen for western blotting to demonstrate that the vortexing process did not cause release of SmCI-1 by damaging of the parasite larvae, and 200 μl of RPMI 1640 (1% Penicillin/Streptomycin) was again added to the transformed schistosomula. Newly transformed schistosomula (NTS) were cultured at 37˚C for 24 hours. Untransformed cercaria, NTS, and NTS excreted/secreted (E/S) products were all gathered for western blot analysis. E/S products were filtered using a 22μm filter prior to storage. An additional experiment was conducted in which schistosomula were cultured for 2 or 4 days, as well as the E/S products released from 0–2 days and 2–4 days, after which samples were collected as previously described, and used in western blotting.

## Western blotting

Parasite samples were resuspended with PBS, to which 4X reducing Laemmli buffer was added, before being boiled at 95˚C for 15 minutes. Samples consisting of 20 cercaria, 20 schistosomula, or the E/S products from 20 NTS were loaded into 12% SDS PAGE gels. Western blotting was performed as per the methods of Hambrook et al. [13], with the exception of 5% BSA being used instead of 5% milk. A 1:1000 dilution of the rabbit anti-SmCI-1 antibody (~0.3μg/ml) was used as a primary antibody. A 1:1000 dilution of a HRP-conjugated goat anti-rabbit polyclonal secondary antibody (Thermo Fisher Scientific) was used as a secondary antibody. Development of the membranes was done using SuperSignal West Dura Extended Duration Substrate (Thermo Fisher Scientific), and chemiluminescent results were visualized using an ImageQuant LAS 4000 Gel Imaging System (GE Healthcare).

## Immunofluorescence staining

Freshly shed cercaria were fixed using 4% paraformaldehyde for 1 hour at room temperature in 1.5 ml tubes. Fixed cercaria were washed 2X using 1000 μl phosphate buffered saline + 0.1% Tween-20 (PBS-T). Permeabilization was performed using 500 μl of PBS + 5%BSA, 0.02% $NaN_3$, 1% TritonX-100 for 1 hour at room temperature. Cercaria were then washed 2X as described above. Parasites were blocked for 1 hour at room temperature in PBS +5% BSA after which a 1:400 dilution of rabbit anti-SmCI-1 (0.8μg/ml) was added for 1 hour. Samples were washed 2X 20 minutes in PBS-T on a rocking platform, and then stained with a 1:400 dilution of donkey anti-rabbit Alexa 568 antibody and 10 μl of Alexa Fluor 488 Phalloidin (Thermo Fisher Scientific) at 4˚C overnight. Samples were again washed 2X 20 minutes with 1000 μl PBS-T and 2X 20 minutes with 1000 μl PBS. Finally, samples were resuspended in 100 μl of PBS, to which one drop of Fluoroshield with DAPI (Millipore Sigma) was added. Fluorescence was visualized using a Leica TCS SP5 laser scanning confocal microscope before analysis via ImageJ (NIH).

## Generic MMP assay

Generic MMP function was determined using a Generic MMP Activity Kit (AnaSpec, AS-72202) as per the kit's instructions. Recombinant SmCI-1 and rSmCI-1Mut at concentrations of 0.25, 0.5, and 1.0 μg/ml were added to a kit specific reaction buffer (KSRB) and were either activated via incubation with 1mM 4-Aminophenylmercuric acetate (APMA) for one hour at 37˚C or left at 37˚C without activation. Inhibition was assessed via the addition of 250 μM of 1,10-phenanthroline (Sigma Aldrich). As a positive control, recombinant human MMP-8 (R&D Systems) was subjected to the same conditions. Substrate only, buffer only, and APMA

controls were also added. 45 μl samples were then mixed with 45 μl of substrate (Mca/Dnp fluorescence resonance energy transfer peptide) in a clear-bottom black-welled 96 well plate and 330nm/390nm fluorescence readings were obtained after 1 hour using a SpectraMax M2 microplate reader (Molecular Devices). Differences between treatments were assessed via One-Way Anova.

An additional experiment was run to determine the activity levels of rSmCI-1 in various buffers of relevance to other assays. Generic MMP activity in KSRB, phosphate buffered saline (PBS, pH 7.4), RPMI 1640 (Gibco), DMEM (Gibco), Krebs-Ringer phosphate Buffer (KRPG, pH7.4), and MMP buffer (50mM Tris, 10mM $CaCl_2$, 150mM NaCl, pH 7.5) was examined after 3 hours of incubation at 37˚C.

### Gelatin, collagen and elastin cleavage assay

Where possible, the capacity of rSmCI-1 to cleave structural molecules was assessed using commercially available kits. Due to the variability of activity between buffers, we elected to use the KSRB from the Generic MMP Activity Kit for these assays. All MMPs were activated via incubation with 1mM APMA as previously described. Gelatin and collagen type 4 cleavage were examined using an EnzChek Gelatinase/Collagenase Assay Kit (Thermo Fisher Scientific), with MMP-2 (AnaSpec) and *Clostridium histolyticum* collagenase (Thermo Fisher Scientific) used as positive controls. Proteases were incubated with 100 μg/ml fluorescein labeled substrate in a clear-bottom black-welled 96 well plate and 495nm/515nm fluorescence readings were obtained after 20 hours incubations at 37˚C using a SpectraMax M2 microplate reader (Molecular Devices). Differences between treatments were assessed via One-Way Anova.

Elastin cleavage was examined using a SensoLyte Green Elastase Assay Kit (AnaSpec). Porcine elastase (AnaSpec) and MMP-12 (R&D Systems) were used as positive controls. Cleavage of the 5-FAM/QXL 520 labeled elastin was measured in a clear-bottom black-welled 96 well plate and 490nm/520nm fluorescence readings were obtained after 1 hour at 37˚C using a SpectraMax M2 microplate reader (Molecular Devices). Differences between treatments were assessed via One-Way Anova (Prism 9; GraphPad Software).

### Silver stain cleavage assays

For relevant structural and immunological proteins for which no relevant cleavage assay was found, silver staining was used to determine the capacity of rSmCI-1 to cleave such molecules. Two μg of Human Fibrinogen (Sigma-Aldrich), Immunoglobulin G (Sigma-Aldrich), recombinant complement component C3 (Millipore Sigma), recombinant CD4 (Sigma Aldrich) were added to 20 μl of KSRB with 1mM APMA. Samples were then either left without protease or given 2 μg/ml rSmCI-1, or 2 μg/ml Trypsin as a positive control for cleavage. Samples were incubated at 37˚C for 18 hours, before being treated with reducing Laemmli buffer, boiled at 95˚C for 15 minutes, and run on 12% and 15% SDS PAGE gels using a Mini PROTEAN Tetra system (Bio-Rad) at 150 V and 180 mA. Silver staining was performed using a Pierce Silver Stain Kit (Thermo Fisher Scientific) and images were obtained using an ImageQuant LAS 4000 Gel Imaging System (GE Healthcare).

### Testing hemolytic activity of human sera pre-treated with rSmCI-1 and rSm-CI-1Mut

Human sera (Sigma Millipore; S7023) was preincubated with rSmCI-1 or rSmCI-1Mut at a concentration of 2μg/ml for 18 hours. Using pre-sensitized sheep erythrocytes(ssRBC) (Comp-Tech, USA; B202), classical pathway mediated hemolysis was assessed following previously

published protocols, with invadolysin treated serum diluted 1:50 and then further diluted 1:2 prior to its addition to a round bottom 96-well plate, prior to the addition of RBCs [62]. Plates were read using spectrophotometer at 412 nm, the cell blank absorbance was subtracted from each measurement to obtain correct absorbances. Fractional hemolysis in each well was calculated relative to the 100% lysis wells.

Alternative pathway-mediated hemolysis was assessed using rabbit erythrocytes (Innovative Research, USA; IRBRBC10ML) following a previously published protocol [62]. Briefly, a 1:25 dilution of the rSmCI-1 and rSmCI-1Mut preincubated sera prepared as described above was prepared and then diluted 1:3 to create a working solution that was added to the wells of a round-bottom 96-well plate, prior to the addition of RBCs. Plates were read using spectrophotometer at 412 nm. Fractional hemolysis in each well was calculated relative to the 100% lysis wells.

## RNAi-mediated knock down of SmCI-1 in cercaria

Recent advances in RNA mediated interference have demonstrated that long dsRNA is taken up into cells via Class A scavenger receptors [63]. Once internalized, these long dsRNA strands are processed into siRNA by Dicer and loaded into the RNA-induced silencing complex (RISC) which can then be used to silence gene expression [64]. In order to utilize long dsRNA to silence expression of SmCI-1 in *S. mansoni* cercaria, we injected 10 *S. mansoni* infected *B. glabrata* with 6nM of dsRNA with repeated sequences complementary to portions of the SmCI-1 transcript (S1 Table). From 7 to 12 days post infection, cercaria were shed from snails as described above, and 20 cercaria were collected for western blot analysis. Membranes were probed with the α-SmCI-1 antibody or using a commercially available α-actin antibody (Thermo Fisher Scientific, 0.5μg/ml). A GFP KD dsRNA construct was used at an identical concentration as a dsRNA control for mouse infection assays (S1 Table).

## Plasma-mediated killing of newly transformed schistosomula

Cercaria were isolated from snails injected with SmCI-1 dsRNA both 12 days post injection (SmCI-1 KD cercaria) as well as after only 8 days post injection as a SmCI-1 positive control. Cercaria were mechanically transformed as previously described and added to the wells of a 24 well plate in 100μl of DMEM at a concentration of 5 cercaria per well. Newly transformed schistosomula were exposed to 10% human serum, with or without the wild type or mutant form of rSmCI-1 at 2μg/ml, or 10% heat inactivated human serum. Larval parasite viability was measured using the previously established method published by Frahm et al., taking into account parasite motility, morphology and granularity [65]. Briefly, immobile, and highly granular parasites with non-intact teguments received a score of 0, while contracting parasites with no granulation and a smooth tegument were given a score of 3. Scoring was performed by a single individual, blind of the treatment.

## Cytokine assessment

Initial exploratory assessments of the effect of rSmCI-1 on cytokine production were completed using THP-1 cells, Jurkat cells, and a combination of both Jurkat and THP-1 cells at a ratio of 3:1. Cells were cultured in RPMI, at 10-cells per treatment. THP-1 differentiation was performed as via treatment with 40nm Phorbol 12-myristate 13-acetate (PMA) for 24 hours followed by a 24 hour rest period in RPMI. Cells were either left unstimulated, stimulated with 225 ng whole cercarial lysate (WCL), 150 ng/ml drained cercarial lysate (DCL), or 75 ng/ml of E/S products. WCL had been generated via 3x 15s sonication of freshly emerged cercaria in PBS, on ice, while DCL and E/S products were obtained in an identical manner after 24 hour

culture of the larvae at 37˚C. Protein concentration was determined via Pierce BCA assay (Thermo Fisher Scientific). The effects rSmCI-1 on cytokine production were assessed by the addition of 2 μg/ml of either rSmCI-1 or rSmCI-1Mut 30 minutes prior to stimulation with *S. mansoni*-derived factors. Cytokine levels were measured 24 hours post stimulation using the Human Proteome Profiler Human Cytokine Array Kit (Bio-Techne/R&D systems). Analysis was performed by comparing the amount of signal produced relative to media-only treated cells.

Investigations into the effect of rSmCI-1 and rSmCI-1Mut on cytokine production from human blood mononuclear cells (PBMCs) were examined using a Human XL Cytokine Luminex Performance Assay 46-plex Fixed Panel (Bio-Techne/R&D systems). PBMCs (Cedarlane/ATCC) were cultured in RPMI at 10,000 cells per treatment. Cells were either left unstimulated, stimulated with *S. mansoni* materials as described above, or stimulated with 1 μg/ml of lipopolysaccharide (LPS). The effects of the recombinants on cytokine production were assessed by the addition of rSmCI-1 or rSmCI-1Mut at a concentration of 2 μg/ml 30 minutes prior to stimulation. Cytokine levels were measured 24 hours post stimulation.

### Western blot cytokine cleavage assay

Recombinant human Eotaxin-1 (R&D systems) and recombinant IL-5 (R&D systems) at an amount of 50 ng/sample were incubated overnight in kit specific reaction buffer with 1 μg/ml rSmCI-1, 1 μg/ml rSmCI-1Mut, or 1 μg/ml trypsin. Western blotting was performed as detailed above, using a 1:1000 (0.3 μg/ml) dilution of monoclonal mouse anti-Eotaxin-1 or IL-5 (R&D).

### Infection and perfusion of mice

Swiss Webster mice were used in infection assays to determine the role of SmCI-1 in *in vivo* infections. Infections were performed using a previously established method during which mice were anesthetized with sodium pentobarbite, after which a suspension of 100 cercaria in 750 μl of water was placed in a plastic ring (1 cm in diameter) on the surface of the mouse for 1 hour [22]. Cercaria treatments included parasites 12 dpi from PBS injected snails, 12 dpi GFP KD controls, 8 dpi SmCI-1 KD larvae, and 12 dpi SmCI-1 KD larvae.

Five weeks post infection, mice were perfused via portal veins and adult worms were quantified using previously established methods [66,67]. Briefly, mice were euthanized using a euthanasia solution containing heparin sodium salt. They were then perfused with a perfusion solution (0.85% NaCl, 0.75% $Na_3C_6H_5O_7$) using a 20-gauge needle inserted into the descending aorta. Adult worms were collected from a small slit in the hepatic portal vein, and counted via stereomicroscope.

### Statistical analysis

Differences in activity in all cleavage assays, cytokine levels, and adult worm recovery each assessed via One-Way Anova (Prism 9; GraphPad Software). Given the non-parametric nature of viability scores, difference between groups in the schistosomula survival assay were assessed using Kruskal-Wallace test at each timepoint, followed by Dunn's multiple comparison tests.

## Results

### Location and expulsion of SmCI-1

Western blot analysis of SmCI-1 released during transformation into NTS demonstrated both the presence of SmCI-1 in cercaria as well as its release during transformation (Fig 1A). Our

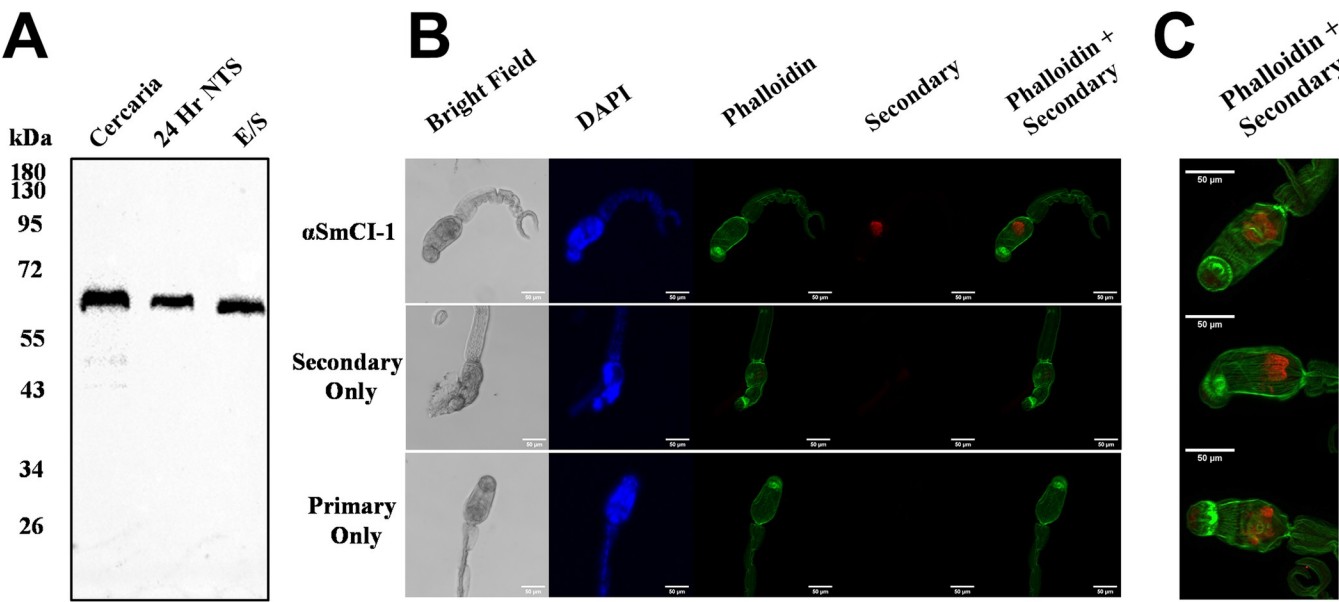

**Fig 1.** (A) Western blot showing SmCI-1 immunoreactivity as a ~65kda band in cercaria, 24 hour post transformation cercaria, and cercarial E/S products. (B) SmCI-1 is detected within the *S. mansoni* cercaria, in an area consistent with localization within the acetabular glands, as revealed by immunofluorescence imaging, with SmCI-1 signal appearing in red. (C) Additional immunofluorescence images feature varying levels of fluorescence with consistent localization within the cercaria.

antibody recognizes a protein appearing at the expected size of 65 kda. The 65 kda band was observed in untransformed cercaria, as well as 24 hr NTS and their E/S products. SmCI-1 protein levels decreased over time as schistosomula developed (S2 Fig). Immunofluorescence staining utilizing a polyclonal anti-SmCI-1 antibody revealed a concentrated fluorescence signal emanating from the head of the cercaria, consistent with localization in acetabular glands (Fig 1B). Staining with either primary or secondary antibodies only failed to elicit such a signal. The SmCI-1 antibody repeatedly recognized a similar region in cercaria consistent with localization in acetabular glands (Fig 1C).

## SmCI-1 displays canonical features and activity of an MMP

SmCI-1 possess the canonical MMP-active site amino acids necessary for function as a matrix metalloprotease (S1 Fig). Recombinant SmCI-1 displays dose-dependent MMP activity and is inhibited via the addition of [250 μM] of 1,10-phenanthroline (Fig 2A). Samples left inactivated, and those of the same concentration that were activated with 1mM APMA did not differ significantly in their activity. The mutation of Glu232→Gly232 rendered the rSmCI-1 mutant completely inactive, with no concentration up to and including 1 μg/ml differing from the vehicle control with respect to general MMP function (Fig 2B). The positive control, human MMP8 also demonstrated MMP activity that was inhibited by the addition of 1,10-phenanthroline (Fig 2C). Unlike rSmCI-1, MMP8 activated with APMA had significantly higher MMP activity than those that were not activated. Among MMP8 samples that were not activated, only those at 1 μg/ml displayed activity levels that differed significantly from the vehicle control (S3 Fig).

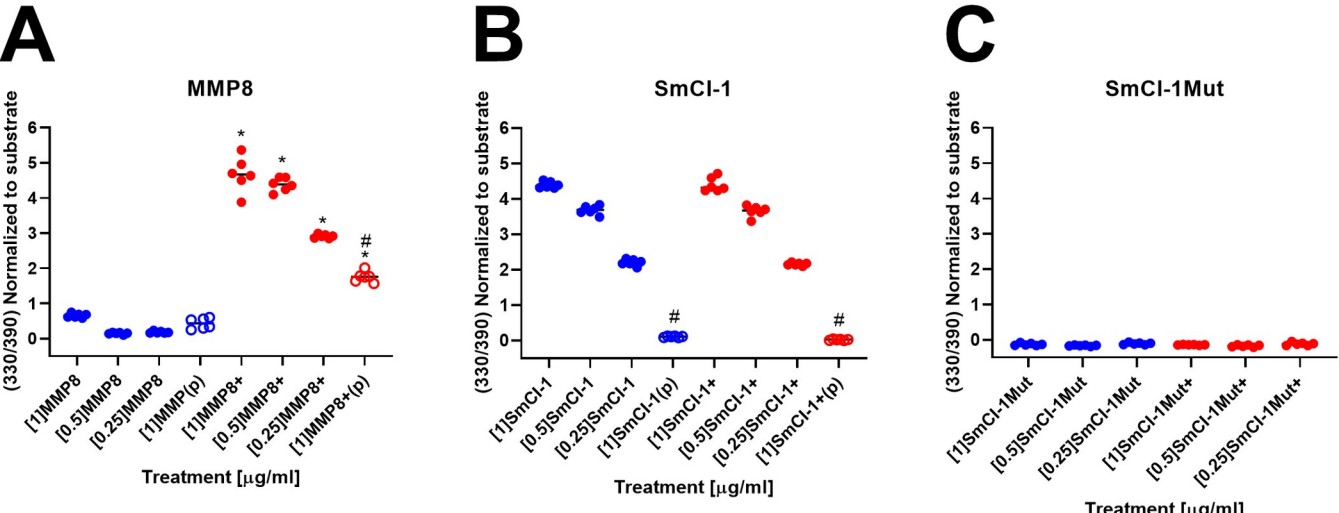

**Fig 2. Recombinant SmCI-1 is a functional matrix metalloprotease.** MMP activity of (A) human MMP8, (B) rSmCI-1, and (C) rSmCI-1Mut. All proteases were examined at three initial concentrations of 1μg/ml, 0.5μg/ml, and 0.25μg/ml. Whereas MMP8 and rSmCI-1 demonstrated activity, rSmCI-1Mut did not. Human MMP8 required activation with 1mM APMA(+), rSmCI-1 had similar activity levels with and without APMA-mediated activation. Significant inhibition via the addition of [50μM] (p) or [250μM] (P) of 1,10-phenanthroline is denoted by (#). Activated samples (+) that differ significantly from their non-activated counterparts of an identical concentration are denoted using (*). (n = 6).

## Cleavage of structural substrates

To better understand the role of SmCI-1 in degradation of prominent components of the extracellular matrix of host skin cells and the basement membrane of human skin, we examined the capacity of SmCI-1 to cleave gelatin, collagen type IV, and elastin. In the case of each of these possible substrates, rSmCI-1 was able to weakly cleave each of them. In the cases of gelatin (Fig 3A) and collagen type IV (Fig 3B), rSmCI-1 cleavage activity significantly differed from the APMA negative control, but only at 2 μg/ml, and not 1 μg/ml. This activity was significantly lower than that observed in the positive control, *Clostridium histolytica* collagenase. In the case of elastin, rSmCI-1 displayed cleavage activity above that of the APMA control for both 2 μg/ml and 1 μg/ml treatments. This activity was again lower than established positive controls for the cleavage of elastin, such as porcine elastase and human MMP12 (Fig 3C).

Under reducing SDS-PAGE conditions, fibrinogen separates into the three distinct subunits of which it is composed (α, β, and γ). Our positive control, trypsin cleaves all three subunits, as evidenced by the disappearance of the bands of all three subunits and the appearance of two bands roughly 40kDa in size. Recombinant SmCI-1, alternatively, cleaves only the alpha subunit of fibrinogen, as shown by the disappearance of the α-subunit band at ~70kDa (Fig 3D).

## Cleavage of complement component C3 leads to increased parasite viability

Four immunologically relevant proteins were examined as possible substrates for SmCI-1: complement component C3, CD4, CR1, and Immunoglobulin G (IgG). The latter three were all cleaved by trypsin but were unaltered by rSmCI-1 treatment (Figs 4 and S4). The only immunologically-relevant protein examined that was affected by rSmCI-1 treatment was complement component C3b (Fig 4A). Trypsin was capable of cleaving both C3b and the C3a anaphylatoxin.

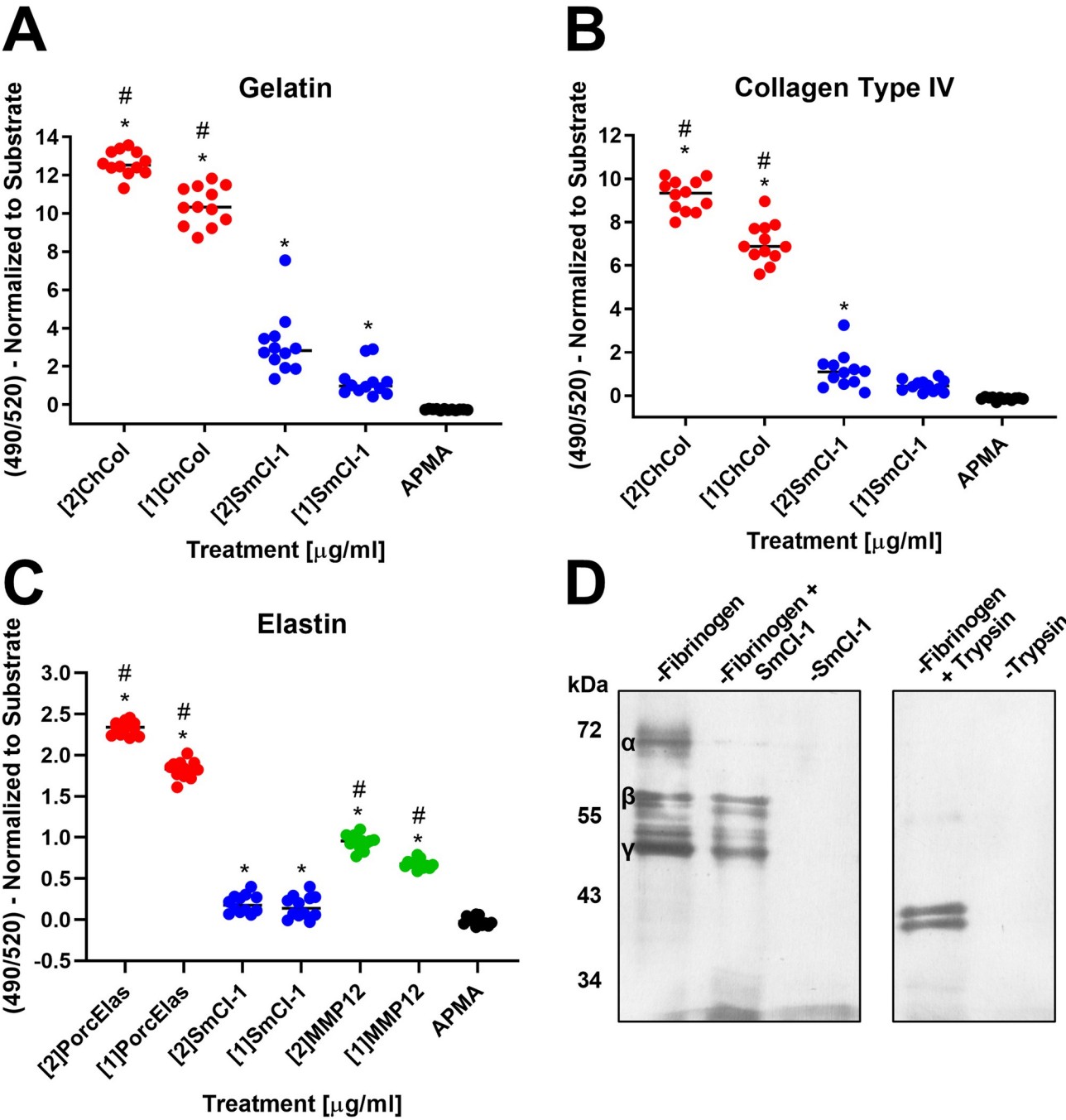

**Fig 3. SmCI-1 weakly cleaves key host ECM components.** Recombinant SmCI-1 cleaves gelatin (A), collagen type IV (B), and elastin (C). In all cases, known positive controls for collagen type 4 and gelatin cleavage (*C. histolytica* collagenase) and elastin (porcine elastase and human MMP12) displayed higher levels of cleavage. Differences from the APMA vehicle control are denoted using (*) and samples with significantly higher cleavage activity than rSmCI-1 are denoted using (#). (D) Silver stain demonstrating the capacity of trypsin to cleave all three subunits of fibrinogen, and that rSmCI-1 can cleave the alpha subunit of fibrinogen only (n = 12).

The cleavage of complement component C3b led us to examine the effects of SmCI-1 on both the classical and alternative complement pathways (Fig 4C and 4D). Pre-treatment of human serum with rSmCI-1 caused a significant decrease in alternative pathway-mediated

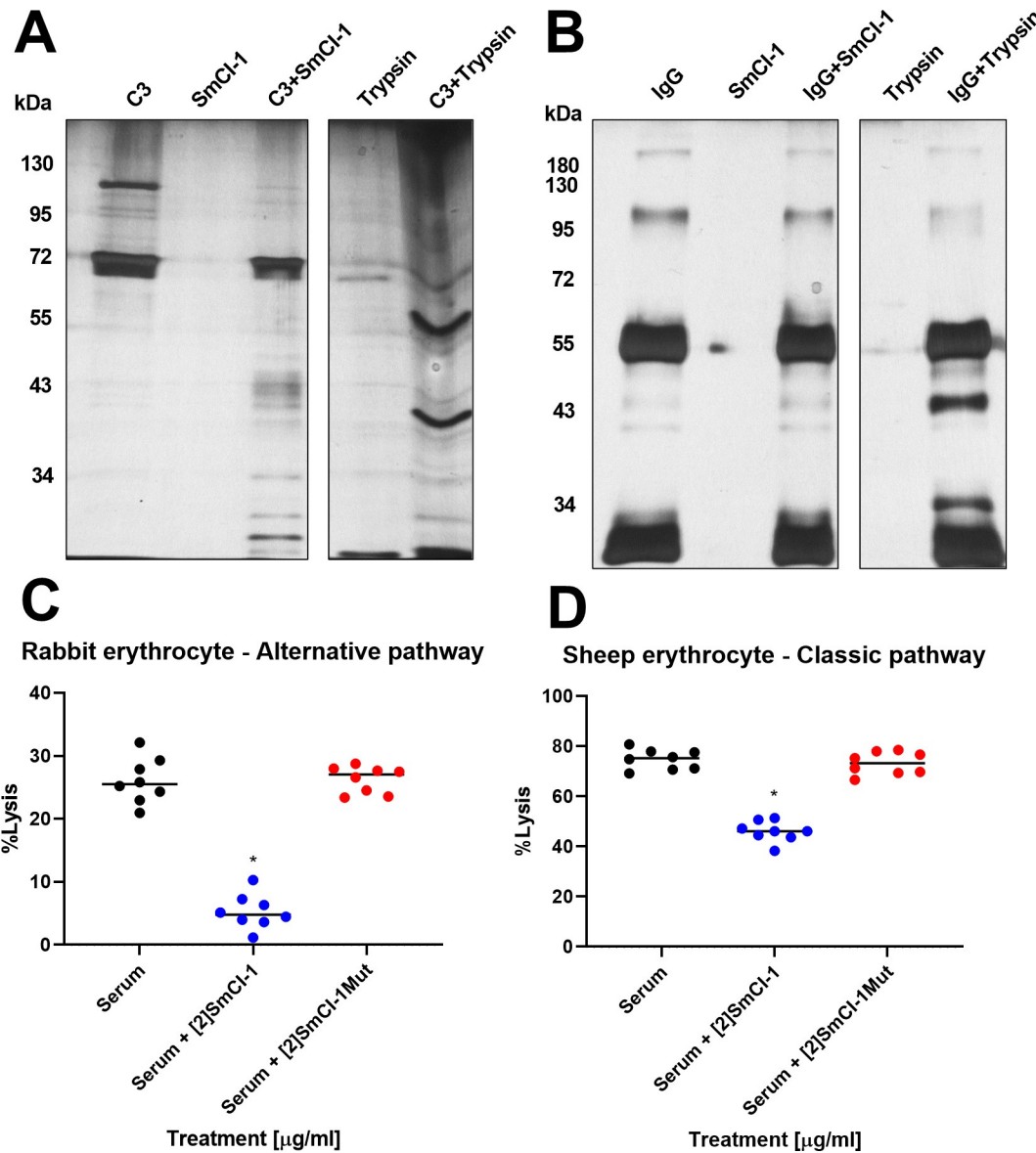

**Fig 4. SmCI-1 Cleaves complement component C3.** Silver stain of 2μg of (A) human complement component C3 loaded into the lanes of a reducing SDS-Page Gel. Under such conditions, C3 is separated into a C3b band at roughly 110kda, and a C3a band of ~72kda. Trypsin treatment resulted in the degradation of both C3b and C3a, while rSmCI-1 degrades only C3b into fragments visible between 26kda and 55kda. (B) Silver stain of human IgG separated into heavy (~50kda), and light (~25kda) chains, with residual complete antibody at higher molecular weights. (C) Rabbit erythrocyte lysis via the alternative complement pathway and (D) Sheep erythrocyte lysis via the classical complement pathway using human serum is decreased by addition of rSmCI-1, but not rSmCI-1Mut. Significant differences from untreated cells indicated by (*). (n = 8).

lysis from 26.1±1.3% to 5.3±1.0%. A similar observation was made of the classical pathway, which displayed a significant decrease in lysis from 74.8±1.4% to 46.0±1.4% when serum was treated with rSmCI-1. Treatment with rSmCI-1Mut did not result in a significant decrease in lysis, with 26.2±1.6% and 73.2±0.7% lysis seen in the alternative and classical pathway assays, respectively (Fig 4C and 4D).

Injection of the SmCI-1 targeting dsRNA construct was successful at reducing the amount of SmCI-1 protein detectable in *S. mansoni* cercaria. On days 7–10 post injection, isolated

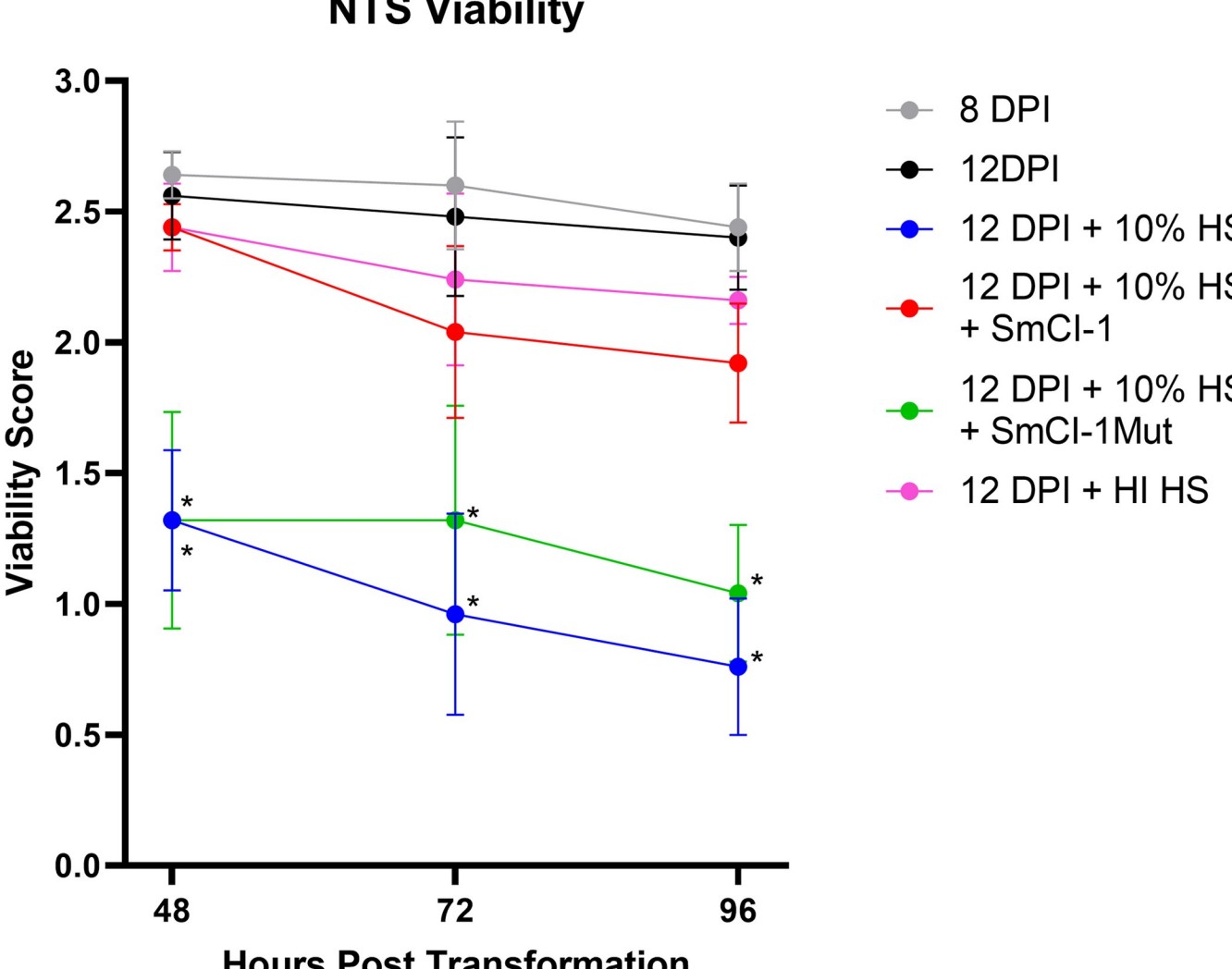

**Fig 5. SmCI-1 KD schistosomula are more susceptible to plasma mediated killing Viability scores of SmCI-1 KD (8dpi) and SmCI-1 KD (12dpi) schistosomula at 2, 3 and 4 days post transformation.** Newly transformed schistosomula were treated with human serum (HS), and either the wild type rSmCI-1, or the catalytically inactive mutant rSmCI-1Mut. Parasites were alternatively treated with heat inactivated human serum (HIHS). (*) indicates where treatments differed significantly from SmCI-1 KD schistosomula exposed to 10% human serum and wild type rSmCI-1. (n = 5).

cercaria were observed possessing SmCI-1, while at 11- and 12-days dpi, SmCI-1 was drastically reduced despite comparable levels of loading control (S5 Fig).

Being able to partially knock down SmCI-1 allowed us to assess the ability of SmCI-1 knockdown (KD) parasites to survive plasma-mediated destruction. Newly transformed schistosomula with SmCI-1 (8 dpi) were as viable as SmCI-1 KD parasites (12 dpi) at 48, 72, and 96 hours post transformation (Fig 5). At all three of the timepoints measured, NTS treated with 10% human serum or 10% human serum + rSmCI-1Mut had viability scores significantly lower than those of NTS treated with 10% human serum + rSmCI-1. Additionally, SmCI-1 KD parasites treated with 10% plasma and rSmCI-1 failed to differ in their viability scores from those SmCI-1 KD parasites treated with heat inactivated human serum (Fig 5).

## Cytokine profiles

Initial examinations of the effects of SmCI-1 on cytokine production were conducted in cell lines (S6 Fig). These assessments revealed the capacity of rSmCI-1 to significantly reduce production of IL-1β, IL-12, and TNF-α in THP-1 cells stimulated with either WCL or DCL. It also reduced IL-12 and TNF-α production by Jurkat cells stimulated with cercarial materials. Co-incubation of both THP-1 and Jurkat cells resulted in the production of Eotaxin-1 and IL-5 by WCL and DCL treated cells, and this production was also reduced via addition of rSmCI-1. Contrary to the other cytokines examined, IL-10 was increased via addition of rSmCI-1 in all three types of cell culture. For each of the aforementioned cytokines other than IL-10, DCL was significantly more immunostimulatory than E/S products, while the inverse was true for IL-10.

SmCI-1 was found to have the capacity to alter the cytokine profiles of human PBMCs, as revealed by cytokine array profiling (S2 Table). Amongst this data lie observations on the effect of rSmCI-1 on the production of key cytokines, as well as whether MMP activity is required for these effects (Fig 6). Eotaxin-1, IL-5, IL-1β and IL-12 release from WCL-stimulated cells was significantly reduced following treatment with rSmCI-1, but not rSmCI-1Mut ($p<0.05$). While a trend towards reduced TNF-α abundance in WCL-treated cells was observed after rSmCI-1 treatment, it failed to reach statistical significance. For DCL treated cells, a significant reduction after treatment with rSmCI-1 was observed for IL-5, IL-1β, IL-12, and TNF-α compared to controls, while Eotaxin-l production was not reduced significantly. Again, rSmCI-1Mut failed to cause a reduction in the abundance of these cytokines when applied to DCL stimulated cells. In the case of IL-5, IL-12, and TNF-α, DCL-only treated cells proved to be significantly more pro-inflammatory than WCL-only treated cells. E/S product stimulated cells were less inflammatory for these cytokines that WCL or DCL treated cells, with the exception of IL-1β.

We also examined how rSmCI-1 and rSmCI-1Mut affected the abundance of anti-inflammatory molecules such as IL-10. In the case of IL-10 generated by WCL-stimulated PMBCs, WT and Mut rSmCI-1 treatments failed to differ significantly from each other, although a trend towards higher IL-10 production is observed when the wild type is added (478±17 pg/ml vs 767±329 pg/ml at 24 hours). In DCL-stimulated cells, IL-10 levels were significantly lower than WCL stimulated cells (168±39 pg/ml). DCL-treated cells to which rSmCI-1 was added increased IL-10 output significantly (767±15pg/ml), while rSmCI-1Mut treatment did not result in a significant increased (169±40 pg/ml). Of note is that E/S products elicited the production of 877±55pg/ml IL-10, which was significantly higher than DCL-only treated cells, but not significantly different than WCL-treated cells.

In PBMCs that were not stimulated with *S. mansoni* factors, but only treated with recombinants, rSmCI-1Mut treatment resulted in the increased abundance of numerous cytokines, while rSmCI-1 did not (S2 Table). The only cytokine altered by both rSmCI-1 and rSmCI-1Mut was IL-10, the levels of which increased after 24 hours, with unstimulated levels of IL-10 seen at 54±8.8 pg/ml, increasing significantly to 303±68 pg/ml and 697±82 pg/ml via rSmCI-1 and rSmCI-1Mut treatment, respectively ($p<0.01$).

A similar pattern for cytokine alteration was also observed for pro-inflammatory molecules and IL-10 in the context of LPS stimulated cells at 24 hrs post stimulation (Fig 7). IL-1β production was significantly reduced, with LPS treated cells producing 1107.3 ±144.5 pg/ml, while cells treated with LPS and rSmCI-1 produced only 515.9±41.5 pg/ml. No significant reduction in IL-1β was seen for LPS+rSmCI-1Mut. LPS treatment resulted in higher IL-1α and TNF-α levels than those produced by cells exposed to WCL, with observed levels of 259.1±34.7 pg/ml and 14648.4±1878.8 pg/ml. Both IL-1α and TNF-α

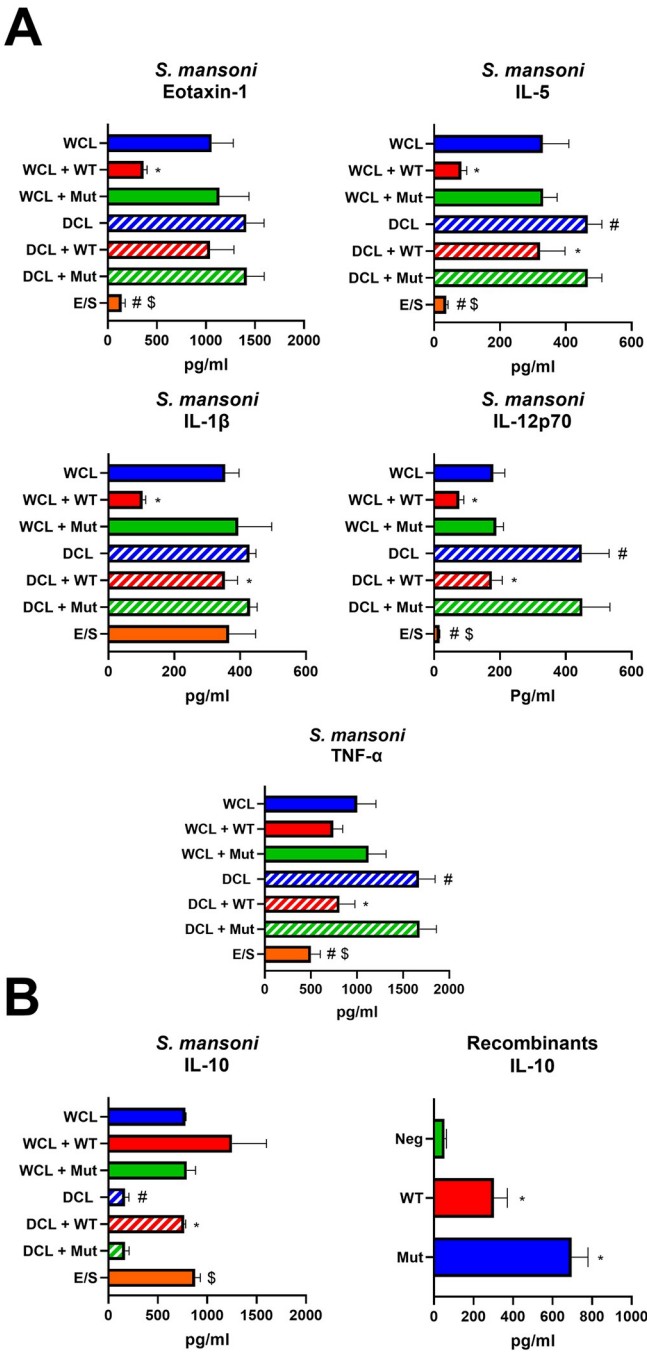

**Fig 6. SmCI-1 alters cytokine profiles in an activity dependent manner when stimulated with whole cercarial lysate.** Cytokine levels from human PBMCs as measured by Luminex panel presented in pg/ml. Differences in levels of key cytokines stimulated with whole cercarial lysate (WCL), drained cercarial lysate (DCL), *or S. mansoni* E/S products. Treatments with 2μg/ml of either wild type rSmCI-1 (WT) or mutant rSmCI-1 (Mut) effects levels of these cytokines. (A) Inflammatory cytokines and key chemokines for which effects are seen. (B) The effect of rSmCI-1 on the anti inflammatory cytokine IL-10 for both *S. mansoni* WCL and DCL-treated cells and cells exposed only to the recombinants. (*) Indicates significant differences between the treatment and either the relevant WCL/DCL control or (in the case of recombinant only treated cells) untreated negative control. (#) Indicates significant difference from WCL treated cells when comparing WCL, DCL and E/S only treated groups, while ($) indicates significant difference from DCL treated cells when comparing levels to E/S product stimulated cells. (n = 3).

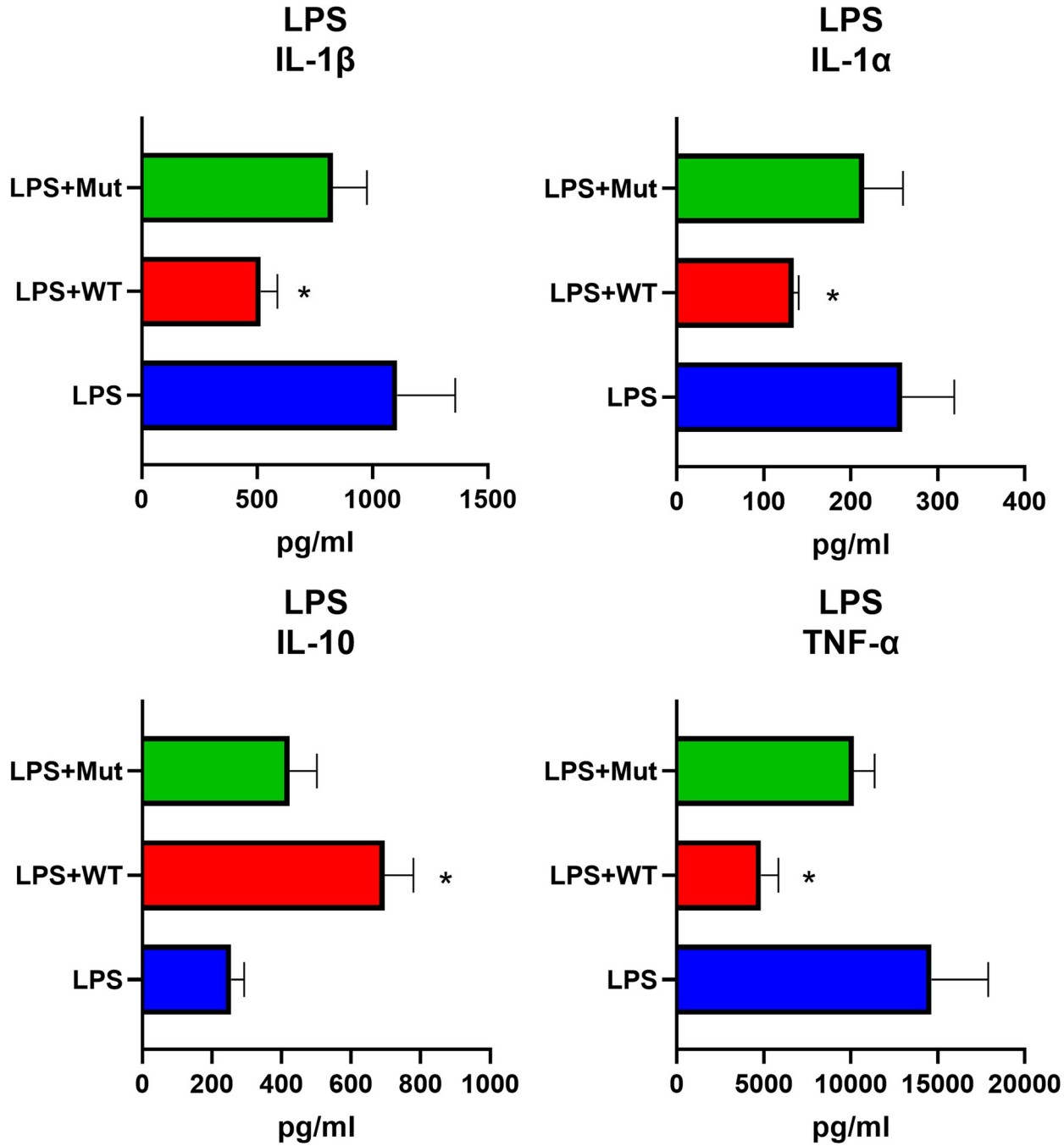

**Fig 7. SmCI-1 alters cytokine profiles in an activity dependent manner when stimulated with lipopolysaccharides.** Cytokine levels from human PBMCs as measured by Luminex panel presented in pg/ml. Differences in levels of key cytokines stimulated with 1μg/ml lipopolysaccharides (LPS). Treatments with 2μg/ml of either wild type rSmCI-1 (WT) or mutant rSmCI-1 (Mut) effects levels of these cytokines. (*) Indicates significant differences between the treatment and either the WCL control or untreated negative control. (n = 3).

levels were significantly reduced via the addition of function rSmCI-1 (p<0.05), but not rSmCI-1Mut. IL-10 increases were also seen when adding either recombinant to LPS stimulated cells. These increases were all statistically significant, with the exception of LPS+rSmCI-1Mut (p = 0.057).

### Cytokine cleavage

Given the capacity of rSmCI-1, but not rSmCI-1Mut, to lower Eotaxin-1 and IL-5 levels, as well as findings in other systems that metalloproteases can cleave Eotaxin-1, we set about examining whether rSmCI-1 is able to cleave Eotaxin-1 and IL-5. Neither recombinant cytokine was cleaved by rSmCI-1 (Fig 8). Trypsin completely degraded Eotaxin-1, and partially degraded IL-5.

### SmCI-1 KD parasites displayed lower infection success in mice

SmCI-1 knockdown was able to significantly reduce adult worm burden 5 weeks post infection in Swiss Webster mice (Fig 9). From the 100 cercaria applied to each mouse, an average of 31.8 (SEM = 2.4) adult *S. mansoni* were recovered from the PBS 12 dpi group, 32.2 (SEM = 1.9) from the GFP KD 12 dpi group, 29.0 (SEM = 2.0) from the SmCI-1 KD 8dpi group, and 22 (SEM = 1.2) from the SmCI-1 KD 12 dpi group. The SmCI-1 KD 12 dpi group featured worm burdens significantly lower than those seen in both the PBS and GFP KD control groups, while the SmCI-1 KD 8 dpi group failed to differ significantly from controls (p<0.01).

## Discussion

Surviving the early stages of infection in human skin, prior to the development of a multi-layered tegument that renders larval schistosomes highly resistant to immune mediated attack, is critical for *S. mansoni* survival [3,68]. We sought to characterize the function of the most abundantly produced invadolysin in *S. mansoni* cercaria [13,23]. We find compelling evidence that SmCI-1 is a potent immunomodulator that can create an environment conducive to survival of the cercaria during its transformation into a schistosomula.

### SmCI-1 is found in *S. mansoni* acetabular glands and expulsed during transformation

The placement of SmCI-1 staining inside of the cercaria is in a location consistent with presence in acetabular glands, which is expected given previous observations in the literature

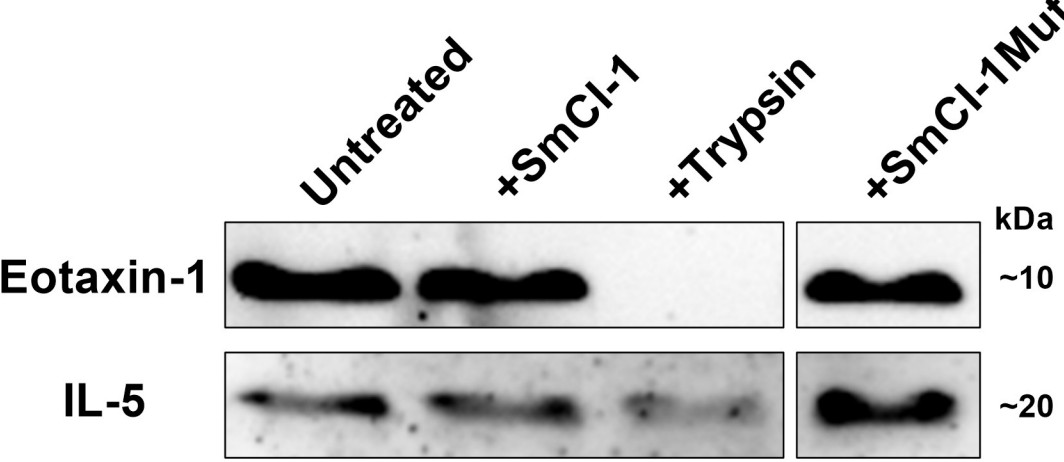

**Fig 8. *S. mansoni* invadolysins do not cleave key cytokines.** Western blots containing 50ng/lane recombinant human cytokines treated with 1μg/ml of the indicated proteases and probed with relevant monoclonal antibodies. Trypsin degrades Eotaxin-1 while the invadolysins do not. IL-5 was weakly affected by trypsin, and not degraded by either the wild type or mutant rSmCI-1.

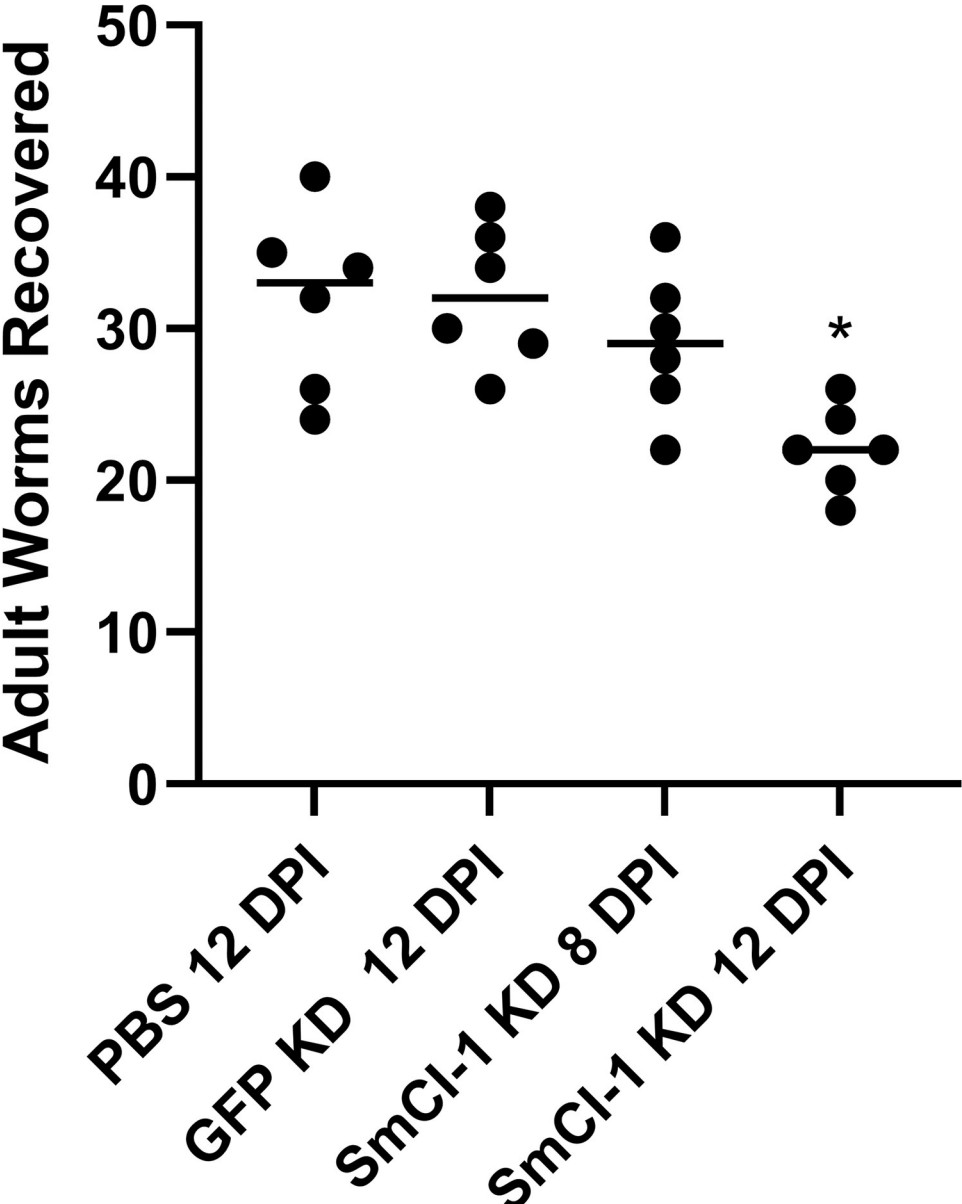

**Fig 9. SmCI-1 KD reduces adult worm burden in mouse model.** Infection of Swiss Webster mice using 100 *S. mansoni* cercaria results in different adult worm burdens 5 weeks post infection depending upon cercaria treatment. Cercaria emerged from *B. glabrata* 12 days post injection with a PBS control, a GFP KD dsRNA construct, a SmCI-1 KD dsRNA construct, or 8 days post injection with a SmCI-1 KD dsRNA construct. (*) Indicates a significantly lowered adult worm burden as compared to PBS 12 dpi and GFR KD 12 dpi groups. (n = 6).

[23,42]. The original identification of SmCI-1 suggested it is a prominent protein of the cercarial acetabular glands (23). Additionally, work by Vondráček et al. using laser microdissection of *Trichobilharzia szidati* revealed the presence of invadolysins in both post and pre-acetabular

gland [42]. However, we were unable to ascertain the precise acetabular gland that SmCI-1 is contained within [42,69–71].

Our analyses of cercarial bodies and E/S products confirm the release of SmCI-1, which is consistent with the expectation that cercaria expulse acetabular gland contents [23]. We detect the expected ~65kda protein by Western blot along with a faint second band roughly 49kda in size in cercaria samples. Cleavage of the zymogen into an active form is a near universal requirement for MMPs, thus the ~49kda likely reflects the presence of an activated form of SmCI-1 retained within the cercaria [35,72,73]. SmCI-1 is gradually released from the cercaria between 0- and 4-days post transformation. The exact kinetics of acetabular gland release and the location surrounding the parasite in the skin has long been contested [14,15,19–21]. Previous research has demonstrated that acetabular gland contents can be present at the apical end of cercaria as they arrive in the dermis, wherein they are likely to encounter numerous immune cell types [74,75]. Additionally, these glands have been observed to atrophy between 48–72 hours post penetration, meaning they are still present as the parasite combats the immune response in the dermis [76]. The vortexing transformation method used in this study is not ideal for studying acetabular gland release kinetics given that the cercaria are not squeezed as they would be moving between skin cells. This more natural path likely facilitates a more rapid expulsion of E/S products. However, the more controlled context of culturing suggests that as the parasite releases SmCI-1 over time, it does not seem to continue to produce SmCI-1 to replace that which it has already expulsed (S2 Fig). This is consistent with the finding that SmCI-1 is most highly upregulated in the germ ball stage of development, as the cercaria prepares to exit the snail, and not in the emerged cercaria themselves [31].

## MMP activity of SmCI-1 leads to the cleavage of key host proteins

SmCI-1 possesses all of the canonical amino acid residues of an invadolysin, including the three histidines employed in the coordination of a zinc ion (His231 His235 His336), the glutamic acid which acts as a nucleophile (Glu232), the methionine underneath the active site which aids in stabilizing metzincin proteases (Met347), and three cysteines serving as candidates for the "cysteine switch" mechanism of activation (Cys182, Cys194, Cys211) [34,61,77–80]. These residues, as well as a 32.6% amino acid similarity to GP63 from *Leishmania major*, support classification of SmCI-1 as an invadolysin [61].

SmCI-1 clearly displays MMP activity that is inhibited by 1,10-phenanthroline further establishing it as a functional metalloprotease. APMA activation of MMP-8, but not SmCI-1, suggests that the latter exists in a pre-activated state. Possible explanations for this are SmCI-1 being self-activating or activating neighboring SmCI-1 molecules. There is precedent for self-activation, as inhibition of the active site of *Leishmania sp.* GP63 reduces maturation [46]. The concept of neighboring invadolysin molecules being responsible for such activation is less likely, given that addition of activated GP63 to GP63 zymogen does not facilitate increased activation of the zymogen [81]. Finally, mutation of the active site glutamic acid into a non-polar glycine completely abrogated the MMP activity of SmCI-1 given the resulting lack of a nucleophilic side chain [80,82].

Examination of the proteolysis of several host skin structural molecules by SmCI-1 was predicated upon the knowledge that *S. mansoni* products have been known to degrade host ECM at a distance from the invading larval schistosome, as well GP63's ability to degrade host ECM proteins [45,83,84]. We elected to examine elastin and collagen type I V cleavage given their abundance in human skin [85,86]. SmCI-1 was capable of cleaving elastin, gelatin, collagen type IV, albeit at relatively low levels as compared to our a positive controls. As SmCE has been shown to be responsible for the majority of the activity required to enter through human

skin, it is likely that the low elastase/collagenase activity demonstrated by SmCI-1 suggests aiding in movement through human skin may not be the primary function of this MMP [27]. Regarding fibrinogen, while not as abundant in the skin as either collagen or elastin, it may be present at the site of penetration to facilitate wound healing and blood clotting, a function that would likely be negatively impacted by the cleavage of the fibrinogen α-subunit by SmCI-1.

Our discovery that SmCI-1 cleaves complement component C3b aligns with studies of GP63 that demonstrate that it can cleave complement component C3 to help survive complement mediated lysis [48,52]. *S. mansoni* must also contend with complement-mediated killing by human serum, but only during the period in which the parasite is undergoing shedding of the immunostimulatory glycocalyx [87]. The degradation of C3b by SmCI-1 results in decreased complement mediated hemolysis via classical and alternative complement pathways, suggesting SmCI-1 may be employed by cercaria to avoid complement mediated lysis. Further support for a role in combatting the complement system is demonstrated in our NTS viability assay, where SmCI-1 KD parasites are damaged by human serum more so than controls. Given the inability of these parasites to generate typical levels of SmCI-1, and that addition of rSmCI-1 rescues parasite viability scores, we argue that there is compelling evidence that SmCI-1 functions in protecting larval schistosomes from the membrane attack complex. This argument is further supported by the inability of heat inactivated serum to reduce viability scores in the same significant manner as regular serum with or without the catalytically inactive rSmCI-1Mut. This is supported by previous work demonstrating membrane attack complex mediated killing of cercaria by normal human serum, but not heat inactivated serum [88]. The likelihood of complement enhanced killing of skin-stage *S. mansoni* is exacerbated by the presence of human eosinophils and neutrophils, which have displayed schistosome killing activity in vitro, and which is enhanced by the addition of complement factors [89]. SmCI-1 mediated cleavage of C3b would create an environment in which *S. mansoni* has less opsonizing factors to avoid, thereby reducing recognition by monocytes/granulocytes during the sensitive skin-penetration period of its lifecycle. That SmCI-1 does not cleave IgG, CD4, or CR1 is notable given *L. major* GP63 is able to cleave CD4, and schistosomula E/S product are known to cleave IgG. This merely points to a degree of specificity possessed by SmCI-1, as seen with other proteases [49,79,90,91].

## The effect of SmCI-1 on cytokine profiles

In response to *S. mansoni* cercarial penetration of the skin, humans produce a variety of immunostimulatory cytokines with the aim of creating an inflammatory milieu in which invading parasites are destroyed [92]. Schistosomes respond to this inflammation by releasing numerous anti-immune factors [3]. These factors range in form and function, from increasing anti-inflammatory molecules such as IL-10 and reducing production of inflammatory cytokines such as IL-1α and IL-1β, to directly affecting various cell functions [3,93–99].

Cytokine profiling following treatment of PBMCs with recombinant SmCI-1 revealed the alteration of numerous cytokines important for an immune response in the skin. The reduction of IL-1β and IL-12, as well as the first observed reduction of Eotaxin-1 and IL-5 by a cercaria specific factor, suggests both reduced inflammation and reduced recruitment/maturation of eosinophils. While drained cercarial lysate proved more immunostimulatory than whole cercarial lysate, addition of SmCI-1 back into the system in the form of the wild type recombinant resulted in cytokine levels comparable to those seen in cells treated with whole cercarial lysate, which would have contained endogenous SmCI-1. Given the importance of Eotaxin-1 in recruiting eosinophils to the site of infection, the function of IL-5 to activate eosinophils, and the capacity of eosinophils to kill schistosomulae *in vitro*, the downregulation of these cytokines by SmCI-1 likely assists in avoidance of eosinophil mediated destruction [89,100–102].

Precedent in the form of hookworm metalloproteases cleaving Eotaxin-1 led us to believe that direct cleavage of cytokines may have been a mechanism of action, but our Eotaxin-1 and IL-5 cleavage assays suggest this not to be the case [59]. This finding suggests to us that another mechanism may be responsible for the downregulation of these cytokines. Possibilities include the cleavage of surface receptors, the targeting of intracellular signalling molecules or affecting upstream immune stimulating pathways that lead to the production of these cytokines. It is possible that SmCI-1 could cleave cell signalling molecules, given the ability of *L. major* GP63 to cleave MRP and NF-κB, and the recent observations that host macrophages, dendritic cells, and neutrophils are all capable of internalizing schistosome E/S products [74]. Given previous work showing TLR4 signalling through the MyD88 pathway is important in recognizing cercarial tegument, such a pathway serves as a possible target [103].

Regardless of the mechanism by which SmCI-1 enacts its effect, our initial investigations into cytokine production using THP-1 and Jurkat cells as model systems for macrophages and T cells revealed that it is likely to be a generic immunosuppressive mechanism in multiple cell types, and not merely the specific targeting of one subset of immune cell. Given macrophages and T-cells are known to be present at the site of cercarial penetration, they are likely targets for SmCI-1 during infection [3,74].

On top of treating PBMCs with recombinants or whole cercarial lysate, we also examined SmCI-1 ability to alter LPS mediated cytokine profiles. The reduction of IL-1β, IL-1α and TNF-α by the active recombinant suggests that SmCI-1 may be enacting its immunomodulatory role via the targeting of inflammatory pathways common to multiple pathogen types. The field of studying the anti-inflammatory properties of helminth antigens is continuously expanding, and seeks to exploit the hygiene/old friends hypothesis in an attempt to utilize worm antigens as anti-inflammatory factors for the treatment of numerous diseases [104–107]. SmCI-1's ability to reduce LPS mediated inflammation positions it within this growing group of anti-inflammatory worm antigens. While different receptors and signalling pathways could be suggested as possible targets for SmCI-1 interference, the identification of such remains to be performed, and will be a significant focus of our research moving forwards.

Our observation that both the catalytically active and inactive forms of rSmCI-1 upregulated IL-10 production suggests this production is elicited in a non-MMP activity dependent manner. Interestingly, the most significant increase in IL-10 levels was seen during treatment of the PBMCs with only the recombinants, and no other stimulant, suggesting that SmCI-1 promotes production independent of other parasite derived factors. Given the observation that *S. mansoni* E/S products have long been known to elicit IL-10, and that *S. mansoni* infection of IL-10 deficient mice results in increased inflammation and slowed parasite migration, the ability of SmCI-1 to induce its production remains intriguing [98]. Our future work will focus on determining what host factors SmCI-1 is interacting with in order to cause this increase in the production of IL-10.

## Effects on *S. mansoni* mouse infection success

Our dsRNA construct injection marks the first successful knockdown of a *S. mansoni* cercarial factor prior to its emergence from the snail and allowed for the generation of parasites lacking normal levels of SmCI-1. Given the roughly 30.8% reduction in adult worm burden observed as compared to cercaria from PBS injected *B. glabrata*, SmCI-1 has demonstrated it enhances parasite survival. We believe the protective effect of SmCI-1 is likely due to a combination of degrading structural components of mammalian skin, overcoming complement mediated lysis, and downregulating inflammatory responses at the site of infection. All such functions are uniquely tailored to survival by the larval parasite during migration and maturation within the skin. Given that SmCI-1 transcript levels are significantly lower during other life cycle

stages other than germball *S. mansoni*, it is likely that skin-stage parasites are the primary beneficiaries of the invadolysin's protective effects [31].

Interestingly, infected human populations have been shown to possess antibodies to SmCI-1, although at lower levels than measured for numerous other key schistosome antigens such as cathepsin B [108]. It is possible that the inability of humans to mount a sufficient antibody response to this antigen allows the larval parasite to continue to employ the invadolysin in overcoming immune defense mechanism. Immunization against SmCI-1 to increase circulating antibody levels may prove beneficial, not in targeting the parasite itself, but rather in altering the immunological milieu in which the parasite resides, and increasing the likelihood of parasite death in the skin.

## Conclusion

SmCI-1 is the second most abundant acetabular gland protein released by *S. mansoni* cercaria during penetration into human skin. This fact alone merits an examination of its roles during infection. Here, we show that SmCI-1 possesses the hallmark structure and sequences of an

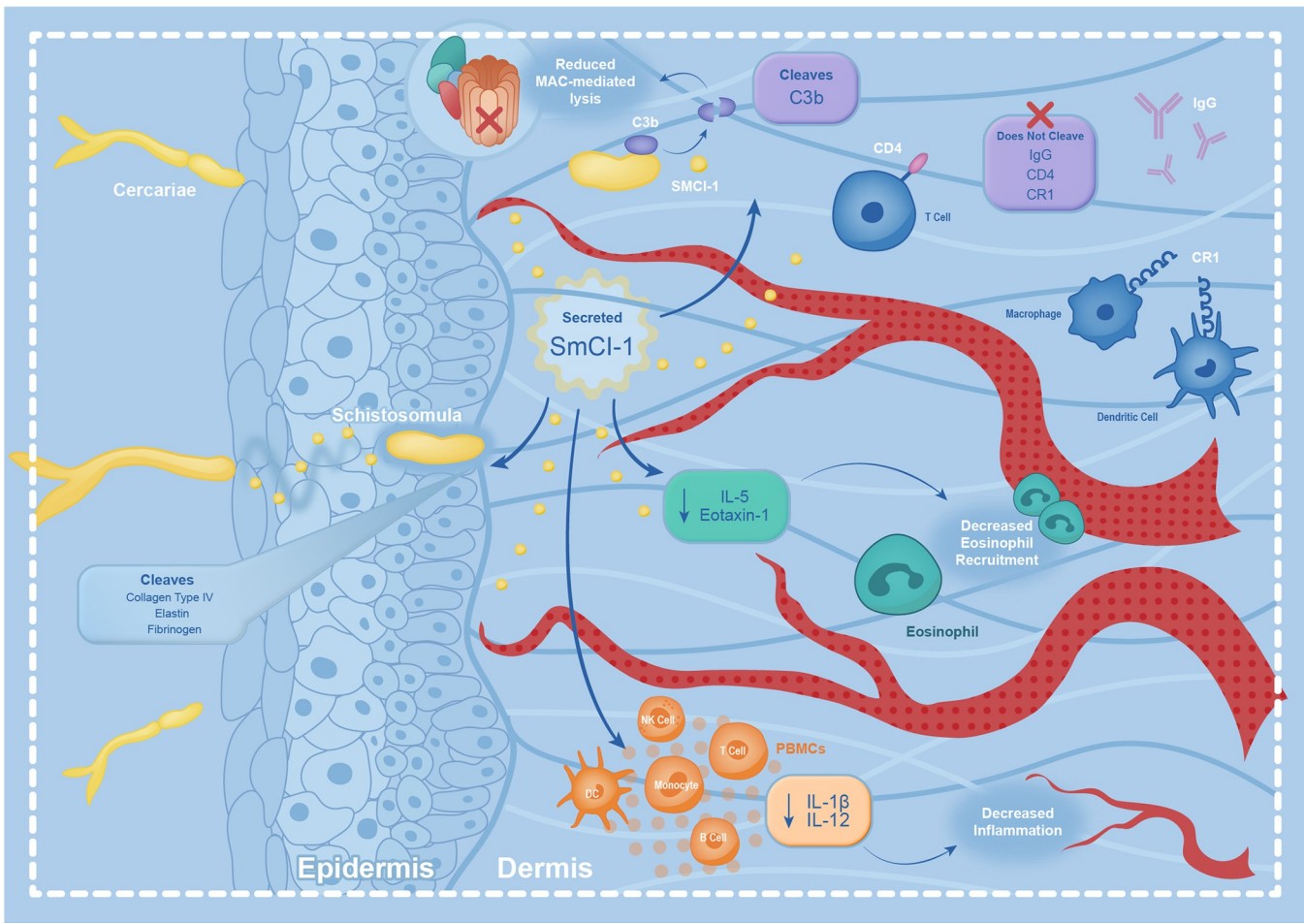

**Fig 10. SmCI-1 functions in numerous ways suggesting a key role in host penetration and survival during the initial stages of intra-human infection.** These include the degradation of host ECM factors involved in the integrity of the skin. It also includes the targeted cleaving of complement component C3b, which leads to the reduction of complement mediated lysis of the parasite. It also includes the reduction of IL-5 and Eotaxin-1 production, an observation suggesting a role in reducing eosinophil recruitment and development. Finally, the reduction of IL-1β and IL-12 caused by SmCI-1 is indicative of a reduction in inflammation in the skin, which might explain the lack of swelling and site-specific inflammation seen during most *S. mansoni* infections.

invadolysin, a family of proteases shown to be important for infection success in numerous parasites. We demonstrate that SmCI-1 has numerous functions relevant to survival in human skin (Fig 10), including cleavage of structural components, targeting of complement component C3b for cleavage, and subsequent reduction of complement pathway activity. This leads to a protective effect against complement mediated lysis of newly transformed cercaria. We also show that it can alter the immunological environment in human cells by reducing IL-1β, IL-12, IL-5, and Eotaxin-1 levels, but only while in possession of MMP activity. Additionally, it can increase IL-10 levels, and MMP activity is not necessary for this function. Finally, knockdown of SmCI-1 in cercaria results in reduced adult worm burdens in mice, confirming the importance of SmCI-1 in infection establishment. Future work should seek to determine the method by which SmCI-1 enacts its effect on cytokines, and seek to determine whether immunization against this factor induces protective immunity against *S. mansoni* infection.

## Supporting information

**S1 Fig.** A 3D visualization of the crystal structure of GP63 (A), and a magnified view of its active site (B). The predicted structure of the full amino acid sequence of SmCI-1 is also present (C) alongside a magnified view of its active site (D). Active site histidines are presented in red, holding a zinc ion (orange ball) in position. Nucleophilic glutamic acids are marked as green, with likely met-turn methionines marked as purple. All cysteines N-terminal to the active site have been coloured yellow.
(TIF)

**S2 Fig. SmCI-1 protein levels in NTS are not continuously renewed.** Western blot of SmCI-1 levels in untransformed cercaria, newly transformed schistosomula (NTS) cultured for two days, and NTS culture for 4 days. Accompanying E/S products obtained from the RPMI cercaria were isolated in, as well as the RPMI supernatant of 0–2 NTS and 2–4 day NTS.
(TIF)

**S3 Fig. SmCI-1 is active in various buffers.** Vehicle, APMA, and 1,10-phenanthroline controls for generic MMP assay fail to vary significantly from substrate controls (A). SmCI-1 activity as measured using a generic fluorometric MMP assay in a variety of buffers (B). The highest activity is seen in the Sensolyte MMP assay provided buffer. Activity is also seen in our lab made generic MMP buffer (50mM Tris, 10mM $CaCl_2$, 150mM NaCl, pH 7.5), as well as RPMI and DMEM. Activity is not seen in Krebs-Ringer Phosphate Buffer (KRPG) (145mM NaCl, 6mM $Na_3PO_4$, 5mM KCl, 0.5mM $CaCl_2$, 1mM $MgSO_4$, pH 7.4) or PBS. Statistically significant differences from vehicle only controls indicated using (\*).
(TIF)

**S4 Fig. SmCI-1.** SmCI-1 fails to cleave CR1 (A), or CD4 (B). Trypsin does cleave these molecules, as evidenced by the appearance of novel bands such as the numerous bands located between 26 and 24kda when it is added to CR1, and the ~20kda bands seen when trypsin is added to CD4.
(TIF)

**S5 Fig. dsRNA successfully knocks down production of SmCI-1.** Western blot using (A) α-SmCI-1 antibody or (B) α-actin antibody on samples of 20 cercaria at varying timepoints post dsRNA construct injection. At 11 and 12 days post injection, SmCI-1 levels are significantly reduced as compared to 7–10 days post injection.
(TIF)

**S6 Fig. SmCI-1 effects cytokine production in both THP-1 and Jurkat cell lines.** rSmCI-1 alters the production of numerous cytokines produced in THP-1 cells, Jurkat cells, and a combination of the two. Stimulation was performed using whole cercarial lysate (WCL, n = 6), drained cercarial lysate (DCL, n = 3), or cercarial excretory/secretory (E/S, n = 3) products. Those samples treated with rSmCI-1 were given 2μg/ml. (*) Indicates a significant difference for rSmCI-1 treated cells as compared to the corresponding WCL- or DCL-only treated samples. (#) Indicates WCL and DCL treated samples with significantly different levels than E/S treated samples. ($) Indicates significantly different levels of production between WCL and DCL treated samples.
(TIF)

**S1 Table. dsRNA sequence used for SmCI-1 KD in cercaria.**
(DOCX)

**S2 Table. Cytokine array data for PBMCs.** Average pg/ml concentrations of individually indicated cytokines treated with *S. mansoni* whole cercarial lysate, E/S products, LPS, recombinant SmCI-1, or recombinant SmCI-1Mut. Values are indicated for both 24 and 12 hour timepoints. Accompanied by Standard error of the mean values (SEM). n = 3.
(XLSX)

## Acknowledgments

Microscopy was performed in the University of Alberta, Faculty of Medicine and Dentistry Cell Imaging Core. The authors would like to recognize Christina Bowhay as the artist and illustrator of the final summary figure.

## Author Contributions

**Conceptualization:** Jacob R. Hambrook, Patrick C. Hanington.

**Formal analysis:** Jacob R. Hambrook.

**Funding acquisition:** Patrick C. Hanington.

**Investigation:** Jacob R. Hambrook.

**Methodology:** Jacob R. Hambrook, Patrick C. Hanington.

**Supervision:** Patrick C. Hanington.

**Writing – original draft:** Jacob R. Hambrook, Patrick C. Hanington.

**Writing – review & editing:** Jacob R. Hambrook, Patrick C. Hanington.

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
