## [Decision Letter · Decision Letter 0]

25 Oct 2022

Dear Dr. Hanington,

Thank you very much for submitting your manuscript "Schistosoma mansoni cercarial invadolysin cleaves complement C3 and suppresses proinflammatory cytokine production" for consideration at PLOS Pathogens. As with all papers reviewed by the journal, your manuscript was reviewed by members of the editorial board and by several independent reviewers. In light of the reviews (below this email), we would like to invite the resubmission of a significantly-revised version that takes into account the reviewers' comments.

Evaluation of this manuscript resulted in 3 divergent recommendations. While all reviewers found merit in the manuscript, significant concerns were also raised by all of them. Given the range of recommendations, I believe it is appropriate to allow the authors the chance to address these concerns, including by adding additional experimental data to bring the manuscript to the standards associated with publication in this journal. Please note that a final decision on the manuscript depends on the ability of the authors to address the concerns of the reviewers; a revised version of the manuscript will be returned to the reviewers for re-evaluation.

We cannot make any decision about publication until we have seen the revised manuscript and your response to the reviewers' comments. Your revised manuscript is also likely to be sent to reviewers for further evaluation.

Sincerely,

Timothy G. Geary, PhD

Guest Editor

PLOS Pathogens

P'ng Loke

Section Editor

PLOS Pathogens

Kasturi Haldar

Editor-in-Chief

PLOS Pathogens

orcid.org/0000-0001-5065-158X

Michael Malim

Editor-in-Chief

PLOS Pathogens

orcid.org/0000-0002-7699-2064

Evaluation of this manuscript resulted in 3 divergent recommendations. While all reviewers found merit in the manuscript, significant concerns were also raised by all of them. Given the range of recommendations, I believe it is appropriate to allow the authors the chance to address these concerns, including by adding additional experimental data to bring the manuscript to the standards associated with publication in this journal. Please note that a final decision on the manuscript depends on the ability of the authors to address the concerns of the reviewers; a revised version of the manuscript will be returned to the reviewers for re-evaluation.

Reviewer's Responses to Questions

**Part I - Summary**

Reviewer #1: The paper describes a good characterisation of an enzyme - a metalloproteinase termed invadolysin - is stored in the acetabular gland of the infective stage of Schistosoma mansoni and based on in vitro experiments is release during the invasion of the host skin. The production of a recombinant form of the enzymes that displays proteolytic activity is well describes and it is shown to degrade various protein stubstrates. The authors also produce a mutant (needs to explain why the particular substitution was chosen) as a control. The degradation studies are performed well as is the substrate studies, although detailed kinetics are not performed. Ii general, this is a well performed characterisation of an important parasite enzyme. There is also data covering the effect of the enzyme on cytokines and on human blood mononuclear cells but these are siply observational studies without follow up functional analysis.

The manuscript needs to be reduced considerably in Introduction and Discussion, and many of the methods are standard and could be written far more succinctly. The weakness is that the paper is a basic characterisation of an enzyme and does not provide anything of particular novelty. Much of the very long Discussion repeats the results and is laboured by lots of speculation but no focused attention of nay new finding.

The title gives the impression that this paper is focused on the complement system but little is done (part froma couple of non-specific assays) that would give any confidence that this protease actually plays a role in neutralising or inactivation this system. Little mention is given to this system in the Introduction, Discussion or Conclusion so why is it the major title?

The effect of the protease on HBMC is interesting but not followed up in detail. Which cell population(s) is targeted.

3D modelling was base on the GP63 structure because they share high homology so is it surprising that the enzyme turns out to have similarities in structure and enzyme activity to this Leishmania molecule.

Many studies are performed on substrate, cytokines, cells but the data in each are preliminary and it's difficult to understand what the authors thinks of all these results and how they fit together - they Conclusion does not make anything more clear apart from stating that the enzyme has many functions displayed in a schematic that is no focus (and certainly does not indicate that complement is a major target for the enzyme).

Reviewer #2: (No Response)

Reviewer #3: This manuscript is original and of importance to researchers in the field of schistosome biology as well as to those interested in host-parasite interactions. As praziquantel is not effective against larval schistosome stages, an increased knowledge of cercarial biology and how they interact with the host is of particular interest in case such research leads to the identification of a cercarial molecule that could be targeted and prevent cercarial survival in the human host. This manuscript represents a comprehensive investigation of the invadolysin SmCI-1, from modelling of its structure to its effects on cytokine release; it represents a substantial amount of experimental work. The study demonstrates the presence of SmCI-1 in cercariae. SmCI-1 demonstrates weak cleavage of ECM proteins but a major finding is that it can cleave C3b suggesting an immunomodulatory role. Furthermore, it can decrease the release of select cytokines when human PBMCs were stimulated with either LPS of whole cercarial lysate.

A few weaknesses of the study are that no marker is provided in the immunocytochemistry examination of SmCI-1 for the acetabulum and/or the acetabular glands. Interestingly the acetabulum did not react with phalloidin whereas it has in a previous study. In addition, whole cercarial lysates (WCL) were used to stimulate cytokine release from human PBMCs that could be reduced by SmCI-1. However, if immunochemistry demonstrates the presence of SmCI-1 I cercariae, the WCL would also include this invadolysin.

The manuscript is of a good quality though requires editing prior to publication. A few comments should also be addressed prior to publication.

**Part II – Major Issues: Key Experiments Required for Acceptance**

Reviewer #1: The paper needs to be much more focused.

1. complement inactivation - discover possible targets for the invadolysin in one or all pathways. The enzyme may degrade 3C in vitro but is there a way to show this in vivo and demonstrate that this is a mean by which the parasite evades the complement system. AS mentioned in the Introduction is release of invadolysin connected with glycocalyx shedding or blocking of 3C binding the the glyoccalyx?

2. The protease may cleave a range of protein in vitro but are these proteins degraded in vivo - could in vivo probing with antibodies show that the parasite secrets the enzyme into the tissues. Does blocking of the enzyme prevent invasion like it is mentioned that blocking the cercarial elastase blocks skin entry.

3. The cellular studies need further exploration - what cells could be targeted by the enzymes and are these relevant to the in vivo situation.

Reviewer #2: See the attached review

Reviewer #3: 1) The acetabulum did not react with phalloidin in the immunochemistry (see Fig 2A) yet it did in “An Atlas for Schistosoma mansoni Organs and Life-Cycle Stages Using Cell Type-Specific Markers and Confocal Microscopy” by Collins et al., 2011. It would be useful to co-stain the cercariae in order to visualize the acetabulum as well as the acetabular glands, or co-stain the acetabular glands with PNA lectin (see Collins et al., 2011). This would support that SmCI-1 immunoreactivity is indeed in the acetabular glands. The authors could then determine whether SmCI-1 is in pre- or post-acetabular glands or both.

**Part III – Minor Issues: Editorial and Data Presentation Modifications**

Reviewer #1: 1. Change title to suit data presented.

2. Shorten Introduction greatly

3. Make Discussion more succinct.

4. Remove modelling studies as no novel information has been gleamed from these.

5. Fig. 4D - please indicate a, b and Y fragments of fibrinogen.

6. Fig. 5 - please add what statistical package was used.

Reviewer #2: (No Response)

Reviewer #3: 2) The naming of samples reacting with the anti-SmCI-1 in western blotting (Fig. 2B) does not necessarily match terminology used lines 177-183. Fig 2B is labelled with cercariae, 24 hour cercariae, and E/S. The labelling needs to correspond more closely to the methods as it is not clear whether the E/S were gathered from transformed schistosomula that had been incubated 24 hours at 37C, or if they are from cercariae immediately post-vortexing.

Also, it would help to be uniform in how to refer to transformed cercariae. It would be helpful to have a timepoint post-vortexing of cercariae that is considered full transformation of schistosomula.

It would be useful to include sample of schistosomula (without supernatant) that have been cultured for 5-6 days to determine that SmCI-1 no longer present in more mature or fully transformed schistosomula.

3) It was very interesting to see decrease in select cytokine release when PBMCs also incubated with whole cercarial lysate (WCL). I am confused by this experimental design. I assume WCL was used to stimulate cytokine release from human PBMCs as this represents cercariae entering the human host. However, WCL was generated by sonicating cercariae 3 x 15 seconds on ice so wouldn’t the WCL (from 5 cercariae) also contain SmCI-1? If so, then the reduction of cytokine release when also incubated with SmCI-1 is due to a high concentration of SmCI-1 that is not physiologically relevant. If WCL does not contain SmCI-1 the authors should clarify that and explain why. Furthermore, the total protein content of WCL should be included.

4) Ref 32 shows that Smp_090100.1 is more abundant in germ balls than cercariae as the authors refer to line 550-2. Might this suggest that SmCI-1 plays a role in emergence from snail? I think the authors should discuss the possible role(s) of SmCI-1 in the snail.

Editing notes

Line 4 if both authors from same institution * not necessary

Line 98 should also cite paper where referred to as Smp_090100.1 (ref 32). If in ref 26 SmCI-1 = SmPepM8 please include this nomenclature

Lines 106-7 should leishmania be Leishmania ?

Line 118 period missing after (54)

Line 148 clarify whether genetic sequence synthesized commercially or via PCR in house

Please include details of company for expression vector

Line 149 Please include details of how transfection performed

Line 153 It would be helpful to state the importance of this particular mutation to function. This is addressed in the conclusions but it would be helpful to include briefly in methods

Please clarify if mutant protein purified the same way as the WT was

Line 155 Please replace “Acetabular genetic sequence” with SmCI-1 for the purposes of uniformity and clarity

Line 158 Kanamycin does not need “K” if not first word of sentence

Please be uniform in use of Thermo Fisher Scientific, varies throughout methods

Line 169 replace “anti-Sm-CI-1” with anti-SmCI-1 for uniformity

Please clarify if antibody was affinity purified using recombinant protein

Line 171 Clarify how many snails were exposed to miracidia

Line 174 Clarify whether snails placed in artificial pond water for cercarial emergence

Line 175 Please include length of vortex step

Line 176 Please include temperature for centrifugation step

Line 179 What will these supernatants be used for?

Line 182 Please clarify whether E/S products were filtered and/or concentrated before use

Line 187 Please clarify quantification of ~ 20 parasites per lane, i.e., does it mean ~ 20 untransformed cercariae, ~ 20 schistosomula, and E/S from ~ 20 schistosomula?

Line 188 Please replace “blotted” with “transferred” and include details of transfer

Line 190 Bovine Serum Albumin does not require uppercase letters

Line 191 Please include the final concentration of the primary antibody in ug/ml

Line 194 Please include company secondary antibody sourced from

Line 201 Please be uniform in use of “cercariae” for plural of cercaria, most other times in manuscript given as cercaria

Line 201 phosphate does not require uppercase p

Line 205 Please include final concentration of antibody in μg/ml

Line 224 Please change “difference” to differences (lines 241 and 247 also)

Line 255 Please include software used for statistical analyses

Lines 228-9 PH should be pH

Line 249 Please change “whom” to which

Line 251 Can abbreviate “micrograms” to μg

Line 254 Why is mutant Sm-CI1 not included? Please be consistent in wording of Sm-CI1 as sometimes comes with “r” in front and other times does not

Line 265 Need punctuation between ingredients of buffer

Line 269-70 please remove “in”

Line 285 M of mM typically not italicized

Line 289 can replace 9g NaCl in 1L with 0.9% NaCl

Line 296 Should bio-techne be Bio-Techne?

Line 307 please insert comma after rMutSm-CI1

Line 308 please include final concentration of antibody used

In controls shown in Fig 2A in methods section

The figure legend for 2B figure states “SmCI-1 is expelled from the cercaria during the initial stages of maturation into schistosomula that occur in human skin”. I would consider this statement a conclusion that should be given in said section rather than in the results. If kept in results a description of the WB should be given first i.e., western blot showing Sm-CI-1 immunoreactivity in…..

Lines 436-8 state “ Eotaxin-1 (1138±301 pg/ml), IL-5 (332±42 437 pg/ml), IL-1β (394±102 pg/ml), and IL-12 (189±21pg/ml) levels all failed to differ from the WCL only treated cells in a significant manner”. Is this line supposed to state that the mutant SmCI-1 does not cause a significant reduction vs WCL stimulation?

Line 442 does “stimmed” mean stimulated? If so please change to latter

Lines 458-9 “The effect of rSmCI-1 on IL-10 levels features an increase is shown for both cells exposed to WCL”. Would change wording as does not flow well, perhaps “is shown” should be omitted.

Line 607 “which have display” should be should have displayed

Line 615 add “was” after “CD4”

Line 623 “factor” should be plural

Line 641 eotaxin starts with lower case e but it is uppercase throughout

Line 646 please give dendritic cells in full

Line 650 “suggest” should be suggests

Line 661 “mediate” should be mediated

Several references contained various formatting issues or lacked italicization of scientific names: 14, 15, 43, 44, 48, 52, 55, 64, 67, 68, 70, 76, 77

For supplementary table 1 please clarify if acetabular gland contents are the E/S from transforming cercariae.

PLOS authors have the option to publish the peer review history of their article (what does this mean?). If published, this will include your full peer review and any attached files.

Reviewer #1: No

Reviewer #2: No

Reviewer #3: No
---

## [Editor Report · Decision Letter 1]

22 Jan 2023

Dear Dr. Hanington,

We are pleased to inform you that your manuscript 'A cercarial invadolysin interferes with the host immune response and facilitates infection establishment of Schistosoma mansoni' has been provisionally accepted for publication in PLOS Pathogens.

Best regards,

Timothy G. Geary, PhD

Guest Editor

PLOS Pathogens

P'ng Loke

Section Editor

PLOS Pathogens

Kasturi Haldar

Editor-in-Chief

PLOS Pathogens

orcid.org/0000-0001-5065-158X

Michael Malim

Editor-in-Chief

PLOS Pathogens

orcid.org/0000-0002-7699-2064

The authors are thanked for the positive and constructive responses to the concerns of the reviewers, some of which were substantial. The reduction in length and increased focus of the Introduction and Discussion are notable. The additional experimental data included in the revision directly and thoroughly address the major concerns raised during the review process. Clear explanations have been provided for why some concerns could not be resolved (e.g., precise localization of the enzyme in the parasite), suggesting that additional experiments are unlikely to be useful in this case. The manuscript provides solid data on a protein important for the process of infection by S. mansoni and its ability to sustain infection in the host. It will be a valuable contribution to the field and to the journal.
---

## [Editor Report · Acceptance letter]

27 Jan 2023

Dear Dr. Hanington,

We are delighted to inform you that your manuscript, "A cercarial invadolysin interferes with the host immune response and facilitates infection establishment of *Schistosoma mansoni*," has been formally accepted for publication in PLOS Pathogens.

Best regards,

Kasturi Haldar

Editor-in-Chief

PLOS Pathogens

orcid.org/0000-0001-5065-158X

Michael Malim

Editor-in-Chief

PLOS Pathogens

orcid.org/0000-0002-7699-2064